# Coupled hydrogeophysical inversion of an artificial infiltration experiment monitored with ground penetrating radar: synthetic demonstration

Rohianuu Moua [1], Nolwenn Lesparre [1], Jean-François Girard [1], Benjamin Belfort [1], François Lehmann [1], and Anis Younes [1]

[1]Université de Strasbourg, CNRS, ENGEES, EOST, ITES UMR 7063, F-67000 Strasbourg, France

**Correspondence:** Rohianuu Moua (rmoua@unistra.fr)

**Abstract.**

In this study, we investigate the use of ground penetrating radar (GPR) time-lapse monitoring of artificial soil infiltration experiments. The aim is to evaluate this protocol in the context of estimating the hydrodynamic unsaturated soil parameter values and their associated uncertainties. The originality of this work is to suggest a statistical parameter estimation approach using MCMC to have direct estimates of the parameter uncertainties. The use of the GPR time data from the moving wetting front only does not provide reliable results. Thus, we propose to use additional information from other types of reflectors to optimize the quality of the parameter estimation. Water movement and electromagnetic wave propagation in the unsaturated zone are modeled using a one-dimensional hydrogeophysical model. The GPR travel time data are analyzed for different reflectors: a moving reflector (the infiltration wetting front) and three fixed reflectors located at different depths in the soil. Global sensitivity analysis (GSA) is employed to assess the influence of the saturated hydraulic conductivity $K_s$, the saturated and residual water contents $\theta_s$ and $\theta_r$, and the Mualem–van Genuchten shape parameters $\alpha$ and $n$ of the soil on the GPR travel time data of the reflectors. Statistical calibration of the soil parameters is then performed using the Markov chain Monte Carlo (MCMC) method. The impact of the type of reflector (moving or fixed) is then evaluated by analyzing the calibrated model parameters and their confidence intervals for different scenarios. GSA results show that the sensitivities of the GPR data to the hydrodynamic soil parameters are different between moving and fixed reflectors, whereas fixed reflectors at various depths have similar sensitivities. $K_s$ has a similar and strong influence on the data of both types of reflectors. Concerning the other parameters, for the wetting front, only $\theta_s$ and $\alpha$ have an influence, and only at long times since the total variance is zero at the very beginning of the experiment. On the other hand, for the fixed reflectors, the total variance is not zero at the very start and the parameters $\theta_s$, $\theta_r$, $\alpha$ and $n$ can have an influence from the very beginning of the infiltration. Results of parameter estimation show that the use of calibration data from the moving or fixed reflectors alone does not allow a good identification of all soil parameters. With the moving reflector, the error between the estimated mean value and the exact target value for $\theta_r$ and $\alpha$ are 9% and 45%, respectively, and less than 3% for the other parameters. The best reduction of the size of the parameter distribution is obtained for $n$, with a posterior distribution 9 times smaller than the prior one. For the other ones, this reduction ratio varies between 1 and 5. For the fixed reflectors, the estimated mean values are far from the target values for $\alpha$, $\theta_r$ and $n$, representing for a reflector located at 120 cm 15%, 27% and 121%, respectively. On the other hand, when both data are

combined, all soil parameters can be well estimated with narrow confidence intervals. For instance, when using both data from the moving wetting front and a fixed reflector located at 120 cm for calibration, the estimated mean values errors of all parameters are less than 5%. Moreover, all parameter distributions are well reduced, with a maximum reduction for $K_s$, leading to a posterior distribution being 46 times smaller than the prior one, and the worst but still satisfactory being for $\theta_r$ for which the posterior distribution is 8 times smaller than the prior one. The methodology was applied to fine, medium and coarse sands with very good results, particularly for the finest soil. The thickness of the unsaturated zone was also tested (0.5 - 1 and 2 m) and a better estimation of the hydrodynamic parameters is obtained when the water table is deeper. In addition, the height of water applied in the infiltrometry test influences the speed of the test without affecting the performance of the proposed method.

**Keywords:** coupled hydrogeophysical model; time-lapse ground penetrating radar; unsaturated soil parameters; global sensitivity analysis; Bayesian parameter estimation; uncertainty quantification.

## 1 Introduction

The vadose zone is defined by the region between the ground surface and the groundwater table. Because of its location, it is at the center of the interactive atmospheric-surface-underground water system. Hence, understanding water flow in the vadose zone is crucial for hydrological modeling and forecasting that can be useful for water resources management, agricultural practices optimization, or geotechnical studies. The porous medium in the vadose zone is filled by both water and air phases. The air phase is considered infinitely mobile and remains at atmospheric pressure. The movement of water has a non-linear behavior and is characterized by two fundamental hydraulic relationships, namely, the water retention and the hydraulic conductivity functions. Various mathematical expressions can describe these functions in terms of dependent variables and fitting parameters. In this work, we use the Mualem-van Genuchten (Mualem 1976, van Genuchten 1980) hydraulic conductivity and water retention models. These models include the following unsaturated soil hydraulic parameters: the saturated hydraulic conductivity, the saturated and residual water contents, and the Mualem–van Genuchten shape parameters $\alpha$ and $n$.

Different approaches can be applied to estimate the unsaturated soil parameters. In soil physics, the reference method relies on laboratory measurements conducted on soil core samples. Such experiments can use various techniques such as thermo-gravimetry or tensiometry, but common practices rely on hydraulic fluxes measurements (Vereecken et al., 2008). Laboratory measurements can provide direct measurements of the soil hydraulic properties or state variables and can therefore be of great accuracy at the column scale. On the other hand, it is prone to certain limitations when the objective is to deduce the soil parameter values at larger scales. Indeed, sample analysis through laboratory experiments is unlikely to provide parameter estimates at field conditions since the volume of the analyzed samples is often not representative of the field heterogeneity at the mesoscale (Scharnagl et al., 2011). In addition, the method is invasive and can be labor-intensive for deep or large scales investigations (Binley et al., 2015). Furthermore, the conservation of collected samples can be challenging because of issues of compaction and changes in porosity.

At the field scale, the soil hydraulic properties and state variables can be estimated using numerous approaches. Measurements of the soil water content, water pressure, and hydraulic conductivity can show significant variations because of their sensitivity to different hydrological processes. As a consequence, such measurements are convenient for the estimation of soil parameters of the subsurface at the field scale by inverse modeling. Soil hydraulic properties and state variables measuring techniques can be classified into two categories, based on whether the measuring devices do have to be in direct contact or not with the soil. In the first instance, when the measuring devices must be in direct contact with the soil, measurements can present a spatial support around the micro (mm - cm) and local scale (cm - m) with water content sensing techniques (using thermal or electromagnetic sensors, e.g., capacitance or time domain reflectometry, Jones et al., 2005; Belfort et al., 2019), water pressure measurements with tensiometers (Cassel and Klute, 1986) or psychometers (Rawlins and Campbell, 1986), and hydraulic conductivity measurements with permeameters (Kodešová et al., 1998) and infiltrometers (Muntz et al., 1905). These techniques can yield data with great resolution at one location and give information on the dynamics at the field scale (Vereecken et al., 2008). In addition, measurements taken at various locations can help to describe the distribution of water content, and thus, allow a good characterization of the state of the soil. For measurements with sensors, however, their installation is often laborious, time-consuming, and destructive (Huisman et al., 2003; Dal Bo et al., 2019). Furthermore, their reliability requires an accurate calibration (Robinson et al., 2008).

Other techniques use non-invasive devices that don't have to be in direct contact with the soil, like remote sensing and hydrogeophysical methods. Remote sensing techniques use devices that operate remotely and relatively far from the ground, such as unmanned aerial vehicles thermal infrared imagery (Zhang et al., 2019) or airborne ground penetrating radar (Edemsky et al., 2021). These methods provide the mapping of water content at a large scale and in locations where contact-based sensing measurements cannot be conducted. However, remote sensing methods exhibit a penetration depth of only a few centimeters and are often limited by the vegetation density (Vereecken et al., 2008; Robinson et al., 2008).

Common hydrogeophysical methods include electromagnetic induction (Doolittle and Brevik, 2014), direct current resistivity (de Jong et al., 2020), nuclear magnetic resonance (Costabel and Günther, 2014), and ground penetrating radar (GPR) (Huisman et al., 2003; Klotzsche et al., 2018) methods. These techniques supply indirect information on hydraulic properties or states, at various scales, from estimated geophysical properties. As mentioned by Binley et al. (2015), such conversion from geophysical to hydraulic properties or states requires the use of robust petrophysical relationships to provide reliable estimates of hydraulic parameters.

Nowadays, GPR is highly used in the field of hydrogeophysics. Different techniques have been reviewed and discussed by Huisman et al. (2003) and Klotzsche et al. (2018). Indeed, GPR is highly sensitive to water content, and, as such, it can close the gap between the spatial scales covered by direct and remote sensing techniques (Klotzsche et al., 2018). Note however that the hydraulic properties estimated from GPR data are subject to an inherent compromise between a deep investigation and a fine spatial resolution. For instance, the lowest frequencies (typically from 1 GHz down to 100 MHz) allow deeper penetrations (until a maximum depth between 1 m and 3 m in most organic media). The temporal variability of the soil water content can be characterized from time-lapse GPR measurements. In this case, the GPR method is applied in a static approach, where, instead of classically imaging the spatial variation of the properties of the subsurface, the device is set immobile and captures how

the properties of the soil change over time. GPR data can be collected during artificial hydraulic processes (e.g., infiltration, runoff, drainage, imbibition) that can provide interesting information on the flow characteristics. Compared to other hydraulic processes, artificially forced infiltration is particularly fast. It also induces a rapidly evolving transient hydraulic perturbation. Time-lapse GPR is characterized by a high spatial and temporal resolution and is therefore well adapted for monitoring such a fast hydraulic process. Artificial infiltration process is also easy to establish since it only requires the application of a positive water pressure head on the soil surface. Hence, time-lapse GPR monitoring of artificial infiltration experiment is usually effortless and time-saving. Furthermore, except in the case of borehole investigations, the GPR device can be laid on or raised above the surface. For these reasons, time-lapse GPR monitoring of artificial infiltration is fast and easy-to-apply and repeat at multiple locations, and, when used on or above the soil surface, non-destructive. Therefore, it is one of the cheapest approach that fits well in the context of mapping the unsaturated soil parameters' heterogeneity at a small catchment scale.

Various studies have investigated the monitoring of different types of flow processes with time-lapse GPR in the context of evaluating the soil hydraulic states, hydraulic properties, or unsaturated soil parameters (e.g., Saintenoy et al., 2008; Moysey, 2010; Scholer et al., 2011; Busch et al., 2013; Tran et al., 2014; Jonard et al., 2015; Jaumann and Roth, 2018; Léger et al., 2014; 2016; 2020).

At the laboratory scale, Léger et al. (2020) have monitored imbibition-drainage experiments using a single-offset surface GPR. Jaumann and Roth (2018) conducted similar experiments but at the test site scale, where they showed reasonable results when estimating the soil unsaturated parameters and the subsurface architecture. As already pointed out however, this hydraulic process can take longer than an infiltration to reach a steady state and is also practically harder to conduct at the field scale. Busch et al. (2013) calibrated the Mualem-van Genuchten parameters of their model by monitoring natural precipitation and evapotranspiration events at the field scale. Such slow hydraulic process can last several days or months and is therefore not suitable for easy and fast characterization. Infiltration experiments have been conducted at the laboratory scale by Moysey (2010), where they considered the GPR two-way travel time (TWT) from various sources of reflection. They showed that the Mualem-van Genuchten shape parameter $n$ is the most poorly constrained among all unsaturated soil parameters. On the field, infiltration processes have been monitored with borehole GPR (Scholer et al., 2011), single-offset (Léger et al., 2014; 2016) and multi-offset (Saito et al., 2018) surface GPR, or off-ground GPR (Jadoon et al., 2008, 2012; Jonard et al., 2015). For practicality, surface GPR is preferred over off-ground and borehole GPR, the latter also being destructive by nature. Saito et al. (2018) used a more complex multi-offset and multi-channel surface GPR to directly monitor the wetting front progression. Mono-channel multi-offset technique is usually not suited for monitoring experiments with high temporal variability, as the offset must be adjusted between each measurement. The multi-channel technique has the advantage to be multi-offset and is, therefore, able to simultaneously determine the propagation speed and the depths of reflectors.

In the present study, we are interested in using a quick, easy-to-apply, and cheap field scale method to characterize the unsaturated soil parameters. To this end, time-lapse GPR monitoring of artificial infiltration is a well suited protocol. It is similar to ring infiltrometry methods but with additional information from GPR measurements. In the literature, the work of Léger et al. (2014) is the closest one considering this protocol for parameter estimation. The authors have demonstrated the relevance of such a methodology to evaluate the hydraulic parameters of sandy soil. They have investigated synthetic and

field examples and showed that the inverted parameters were in agreement with the values obtained in the laboratory for soil samples and with disk infiltrometer measurements. However, in their study, Léger et al. (2014) used an optimization-based inversion algorithm which did not allowed to assess the reliability of the estimated values since the uncertainty associated with the calibrated parameters has not been evaluated. Furthermore, Léger et al. (2014) employed only the TWT data obtained from the GPR reflection on the wetting front for the calibration of the soil parameters, which was satisfactory enough for them to obtain such remarkable results. The original work presented here aims to extend the actual state of the art by:

- Considering different reflectors at different depths: a moving reflector which corresponds to the infiltration dynamic wetting front and two fixed reflectors located at different depths in the soil.

- Investigating the influence of all soil parameters (the saturated hydraulic conductivity, the saturated and residual water contents, and the Mualem–van Genuchten shape parameters $\alpha$ and $n$) on the GPR TWT data of the three reflectors using Global Sensitivity Analysis (GSA). The GSA allows estimating the soil parameters range where time-lapse GPR data monitoring is sensitive to these parameters. It also provides some insight about which parameter is more sensitive at the beginning of the infiltration experiment or at the end of the infiltration.

- Performing statistical calibration of soil parameters using the Markov Chain Monte Carlo (MCMC) method and evaluating the reliability of the estimated parameters by analyzing not only the calibrated model parameters but also their associated uncertainty.

- Evaluating the impact of the type of reflector (moving or fixed) by analyzing the calibrated model parameters and their confidence intervals for different scenarios.

The plan of the paper is as follows: Section 2 describes the test case as well as the mathematical and numerical hydrogeophysical models. Section 3 reports on the GSA results of the different TWT signals. Then, Section 4 discusses the results of soil parameter estimation with MCMC for different scenarios including varying soil types, water table depths and surface boundary conditions.

## 2 Test case description and numerical solution

### 2.1 Test case description

In this work, we conduct a synthetic study on the time-lapse GPR monitoring of artificial infiltration protocol, prior to applying it in real conditions. The idea is to perform synthetic experiments under the same conditions of real experiments to better understand the pertinence of the investigated protocol when used for estimating the unsaturated soil parameters. The test case considered is a hypothetical one-dimensional experiment of water infiltration in a homogeneous sandy soil of 150 cm (Fig.1a). The approach used to drive the artificial infiltration is comparable to other techniques commonly used to estimate the properties of the porous medium, such as single or double ring infiltrometry. As evidenced in other studies (e.g., Léger et

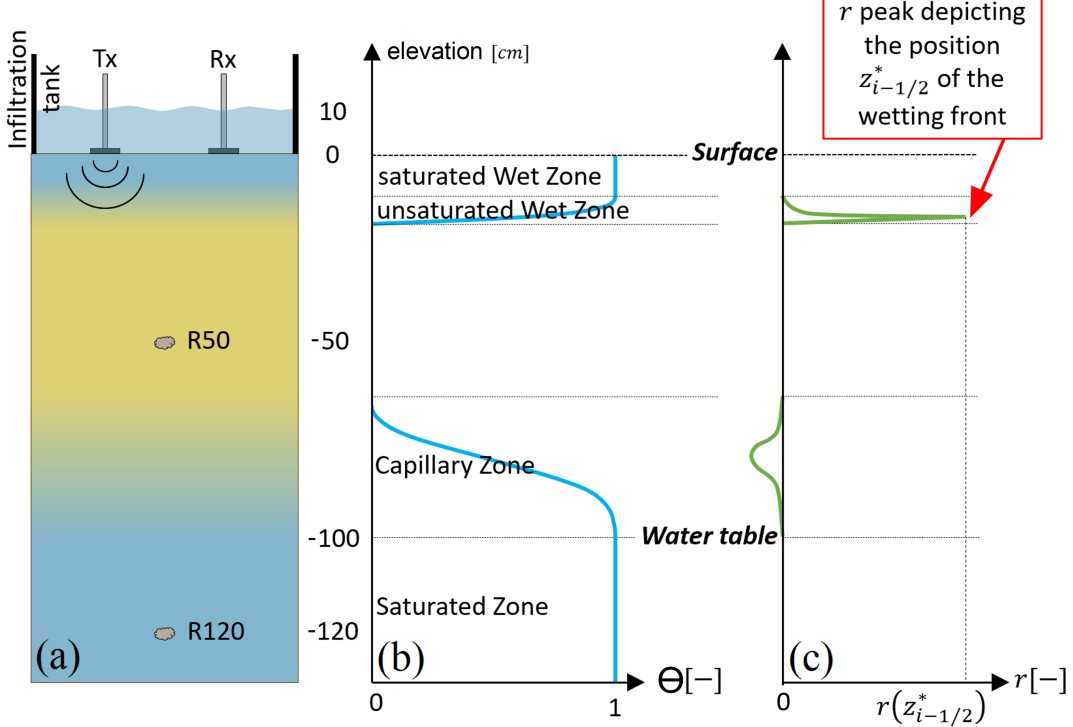

**Figure 1.** Test case and experimental device illustration at an advanced time step (a). R50 and R120 are fixed reflectors considered in this experiment. TX and RX refer to the transmitter and receiver antennas of the GPR system. Effective saturation $S_e$ (b) and reflection coefficient $r$ (c) profiles with depth.

al., 2014), the idea is to add information from the GPR data monitored during the infiltration to have access to more of the hydrodynamic parameters. In the present synthetic case, a constant pressure head of 10 cm is applied at the surface of the soil (i.e., a 10 cm water ponding Dirichlet-type boundary condition is maintained at the top). The medium is initially at the hydrostatic equilibrium with a water table maintained at 100 cm below the soil surface (Fig.1b). The domain is initially formed by an unsaturated zone of 100 cm thick above a saturated zone of 50 cm thick. We assume the experiment to be monitored with a surface GPR. The propagating time (i.e., the TWT) of the GPR waves reflected by two types of reflectors are considered (Fig.1c): (i) the moving infiltration wetting front and (ii) two fixed reflectors corresponding to a local heterogeneity at two different depths. For instance, these can be small objects that are artificially buried (e.g., moisture sensing probes) or naturally embedded (such as small rocks) in the porous medium. The fixed reflectors are supposed to be small enough compared to the section of the infiltrated area, so they do not significantly perturb the vertical flow. The upper fixed reflector, R50, is located in the initially unsaturated zone at 50 cm depth. The reflector R120 is located in the saturated zone, under the water table, at a distance of 120 cm from the soil surface (Fig.1a). In the following, the time-lapse TWT signal for reflection caused by the

infiltration wetting front is noted $TWT_f$ and that from the two immovable diffracting points R50 and R120, are respectively noted $TWT_{50}$ and $TWT_{120}$.

## 2.2    The mathematical model

### 2.2.1    Unsaturated flow model

Water infiltration in unsaturated/saturated soils is governed by the one-dimensional Richards' equation (Richards, 1931):

$$\frac{\partial \theta}{\partial t} = \frac{\partial}{\partial z}\left[K(\theta)\left(\frac{\partial h}{\partial z} - 1\right)\right] \tag{1}$$

where $h$ (cm) is the pressure head; $z$ is the depth (cm), taken positive in the downward direction; $t$ is the time (s), $\theta$ (cm$^3$/cm$^3$) is the actual water content, and $K(\theta)$ (cm/s) is the hydraulic conductivity which is a function of water content. The initial condition is a hydrostatic pressure distribution corresponding to a water table at 100 cm depth. The boundary condition at the top of the domain is a fixed Dirichlet condition of 10 cm maintained during the experiment. The boundary condition at the bottom is a piezometric head fixed at -100 cm which corresponds to the water table position (Fig.1).

The interdependencies of the pressure head, conductivity, and water content are described using the standard models of Mualem (1976) and van Genuchten (1980):

$$S_e(h) = \frac{\theta(h) - \theta_r}{\theta_s - \theta_r} = \begin{cases} [1 + (\alpha|h|)^n]^{-m} & \text{if } h < 0 \\ 1 & \text{if } h \geq 0 \end{cases} \tag{2}$$

$$K(h) = \begin{cases} K_s \times S_e(h)^L[1 - (1 - S_e(h)^{1/m})^m]^2 & \text{if } h < 0 \\ K_s & \text{if } h \geq 0 \end{cases} \tag{3}$$

where $S_e(h)$ (-) is the effective saturation, $\theta_s$ and $\theta_r$ (cm$^3$/cm$^3$) are the saturated and residual water contents, respectively, $K_s$ (cm/s) is the saturated conductivity, $m = 1 - 1/n$, $\alpha$ (1/cm), $n$ (-) are the Mualem-van Genuchten shape parameters, and $L$ (-) is a parameter characterizing the tortuosity of the flow paths of moving water in the interconnected pores of the soil. It is set at $L = 0.5$ here, following the works of Mualem (1976) and van Genuchten (1980).

### 2.2.2    Petrophysical and Geophysical relationships

In GPR sounding, pulses of radiofrequency (MHz to GHz) electromagnetic waves are emitted from a transmitting antenna through the sounded medium. The electromagnetic response is then acquired with a receiving antenna. With a surface GPR, both antennas are installed at the surface of the soil (Fig.1). To monitor the experiment of water infiltration with time-lapse GPR, the sounding system is set immobile above the infiltration zone in order to capture the time variation of the electromagnetic response due to the change of saturation.

To describe the dependency of the dielectric permittivity on the water content, we use the complex refractive index model (Birchak et al., 1974). This petrophysical relationship relates the dielectric constant $\epsilon$ (-) of a three-phase (water-solid-air) medium to its water content by:

$$\sqrt{\epsilon(z,t)} = \theta(z,t)\sqrt{\epsilon_w} + (1-\phi)\sqrt{\epsilon_s} + [\phi - \theta(z,t)]\sqrt{\epsilon_a} \tag{4}$$

where $\phi$ (-) is the porosity, considered equal to the saturated water content $\theta_s$, $\epsilon_w = 80$, $\epsilon_s = 2.5$ (Léger et al., 2014) and $\epsilon_a = 1$ are the dielectric constants of water, silica (sand) and air, respectively.

In this work, the soil is considered as a linear and isotropic non-magnetic medium. When working with frequencies below 1 GHz, the soil electrical conductivity can be neglected. In this case, the electromagnetic waves propagate at a speed $V$ (cm/ns) (Annan, 2003):

$$V = \frac{c}{\sqrt{\epsilon}} \tag{5}$$

where $c \approx 30$ cm/ns is the speed of electromagnetic waves in air, and $\epsilon$ (-) is the dielectric constant of the porous medium. Equations (4) and (5) evidence that GPR waves propagate at a much lower speed in wet conditions. Any source of reflection in the sounded soil produces a reflected wave that is recorded at a time corresponding to the duration of its propagation, from the transmitting antenna, down to the source of reflection, then back up to the receiving antenna, i.e., the TWT of the reflected wave.

We consider a one-dimensional scenario (the offset between the antennas is null) and discretize the domain into $N$ cells $i$, centered at a depth $z_i$, with element boundaries at $z_{i-1/2}$ and $z_{i+1/2}$. The TWT for the reflection occurring at the interface $(i-1/2)$ between the elements $i-1$ and $i$ can be expressed as the sum of the vertical TWT in each element above $i$:

$$\text{TWT}(z_{i-1/2}) = 2\sum_{j=1}^{i-1} \frac{|l_j|}{V_j} \tag{6}$$

in which $|l_j|$ (cm) is the length of the element $j$ above $i$ and $V_j$ (cm/ns) is the GPR propagation speed in the element $j$.

A reflection occurs at the interface between two successive elements if the reflection coefficient is not zero. The reflection coefficient expresses the contrast of dielectric constant (due to the contrast of water content) at the interface between the two elements $i-1$ and $i$. When the offset between transmitting and receiving antennas is null, the reflection coefficient at interface $(i-1/2)$ is defined by:

$$r(z_{i-1/2}) = \frac{\epsilon(z_i) - \epsilon(z_{i-1})}{\epsilon(z_i) + \epsilon(z_{i-1})} \tag{7}$$

where $\epsilon(z_i)$ is the dielectric constant of the element $i$.

For an 800 MHz antenna, the wavelength can typically vary from 6 cm in a wet medium to around 18 cm in a dry medium. The abrupt change in the reflection coefficient at the wetting front makes it easily detectable. This statement is true in the presented test case and for any parameter value taken from the prior distributions tested (Table 1). On the contrary, the water table may

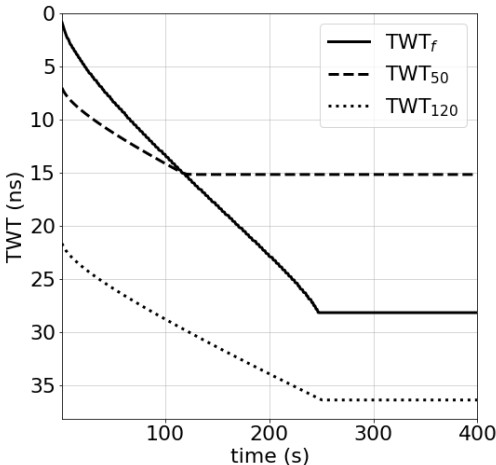

**Figure 2.** Hydrogeophysical model responses for $K_s = 0.08$ cm/s, $\theta_s = 0.4$ (-), $\theta_r = 0.07$ (-), $\alpha = 0.145$ cm$^{-1}$, $n = 2.68$ (-). TWT$_f$ corresponds to the TWT signal for the wetting front, while TWT$_{50}$ and TWT$_{120}$ are the TWT signals for fixed objects located at 50 and 120 cm below the surface, respectively.

be hidden due to the softer change in the reflection coefficient at the capillary fringe (Bano, 2006; Saintenoy and Hopmans, 2011).

Note that one could easily consider a non perpendicular incidence of the GPR wave at the interface, introducing the incidence angle in Eq. 7. Nevertheless, the offset between the TX and RX antenna for a 800 MHz GPR system is typically around 10 cm. By simple trigonometry, the incidence angle is 5 deg at 50 cm depth, 2 deg at the deeper reflector, then the reflection coefficient is very close to the normal incidence, and Eq. 7 is considered in the following. Considering more closely the physics of the radar wave emission and propagation in porous media, if one needs to consider precisely the wave amplitude (such as in full waveform inversion), one should consider the radiation pattern of the antenna. This later is linked to the dielectric contrast at the surface and the antenna characteristics. Whether one needs to calculate it precisely or one should consider specific acquisition configuration to handle this effect (generally normalization of the signal by a reference signal). We choose to keep our approach as simple as possible, to be applicable to any system easily, without the need of calibration. Hence, we have chosen to consider only the travel time, and to not use the amplitude. Note that the later is more sensitive to attenuation (electrical conductivity) than to the hydraulic parameters.

## 2.3 The numerical model

The variation of the water content in the soil during the infiltration is computed using the WAMOS-1D code (Belfort et al., 2018). The model describes the water movement in the porous medium using Richards' equation (1), and the constitutive relationships between the pressure, the hydraulic conductivity, and the volumetric water content given by Eq. (2) and Eq.

(3). The domain of 150 cm depth is discretized with uniform elements of 1 cm thick with homogeneous properties. Such discretization allows an appropriate model precision and a low enough computation time. The WAMOS-1D code solves the system of Eqs. (1)-(3) and yields the vertical distribution of water content at each time step. This distribution is then converted into a vertical dielectric permittivity profile $\epsilon$ using the petrophysical relationship Eq. (4) and into a GPR wave propagation speed profile $V$ using Eq. (5). Then, the time-lapse TWT signals for the fixed objects, $TWT_{50}$ and $TWT_{120}$, are calculated at each time step using Eq. (6) (dashed and dotted curves in Fig.2).

The time-lapse signal $TWT_f$, induced by wave reflection on the wetting front because of the sharp water content variation at the front position is calculated in two steps. First, we search the wetting front position $z^*_{i-1/2}$, which corresponds to the interface position having the maximum reflection coefficient from Eq. (7) as illustrated in Fig.1. Then, the TWT signal of the wetting front is obtained using $TWT_f = TWT(z^*_{i-1/2})$ from Eq. (6) (solid curve Fig.2).

Note that $TWT_{50}$ and $TWT_{120}$ signals are induced by fixed objects, thus, these signals exist regardless of the position of the infiltration front. On the other hand, $TWT_f$ is induced by the infiltration wetting front whose position varies over time. Besides, contrarily to $TWT_{50}$, and $TWT_{120}$, the $TWT_f$ signal disappears when the wetting front reaches the water table. To avoid numerical issues when simulations are performed with different soil parameter sets, the value of $TWT_f$ when the water table is reached, is artificially maintained for the remaining time steps until the end of simulation time. The water table is assumed to be reached when the maximum reflection coefficient of Eq. (7) is under a threshold of $10^{-2}$. This reflects a fully saturated domain with an almost uniform water content distribution (solid curve Fig.2). An explanation of the computation of all TWT signals is summarized in Fig.3.

## 3 Global sensitivity analysis of TWT signals

### 3.1 GSA method

The GSA method evaluates how the outputs of a model are influenced by the variation of the input parameters (Mara and Tarantola, 2008). Among the various forms of GSA, a variance-based sensitivity analysis, allowing the calculation of Sobol sensitivity indices (Sobol', 2001) is employed. Such indices depict the contribution of the variation of any input variable $x$ to the total variance of an output variable $y$. In our case, the input variables are the unsaturated soil parameters $(K_s, \theta_s, \theta_r, \alpha, n)$ and the output variables are the TWT signals ($TWT_f$, $TWT_{50}$, $TWT_{120}$).

Given a model with a set of $p$ independent random parameters $\boldsymbol{X} = \{x_1, x_2, ..., x_p\}$ that yields a random response $y(\boldsymbol{X})$, the two variance-based sensitivity measures, also called Sobol indices (Sobol', 2001) are:

- the first-order sensitivity index:

$$S_i = \frac{\mathrm{Var}\left[E\left[y(\boldsymbol{X})|x_i\right]\right]}{\mathrm{Var}\left[y(\boldsymbol{X})\right]} \in [0, 1] \tag{8}$$

- the total sensitivity index:

$$ST_i = \frac{E\left[\mathrm{Var}\left[y(\boldsymbol{X})|x_{-i}\right]\right]}{\mathrm{Var}\left[y(\boldsymbol{X})\right]} \in [0, 1] \tag{9}$$

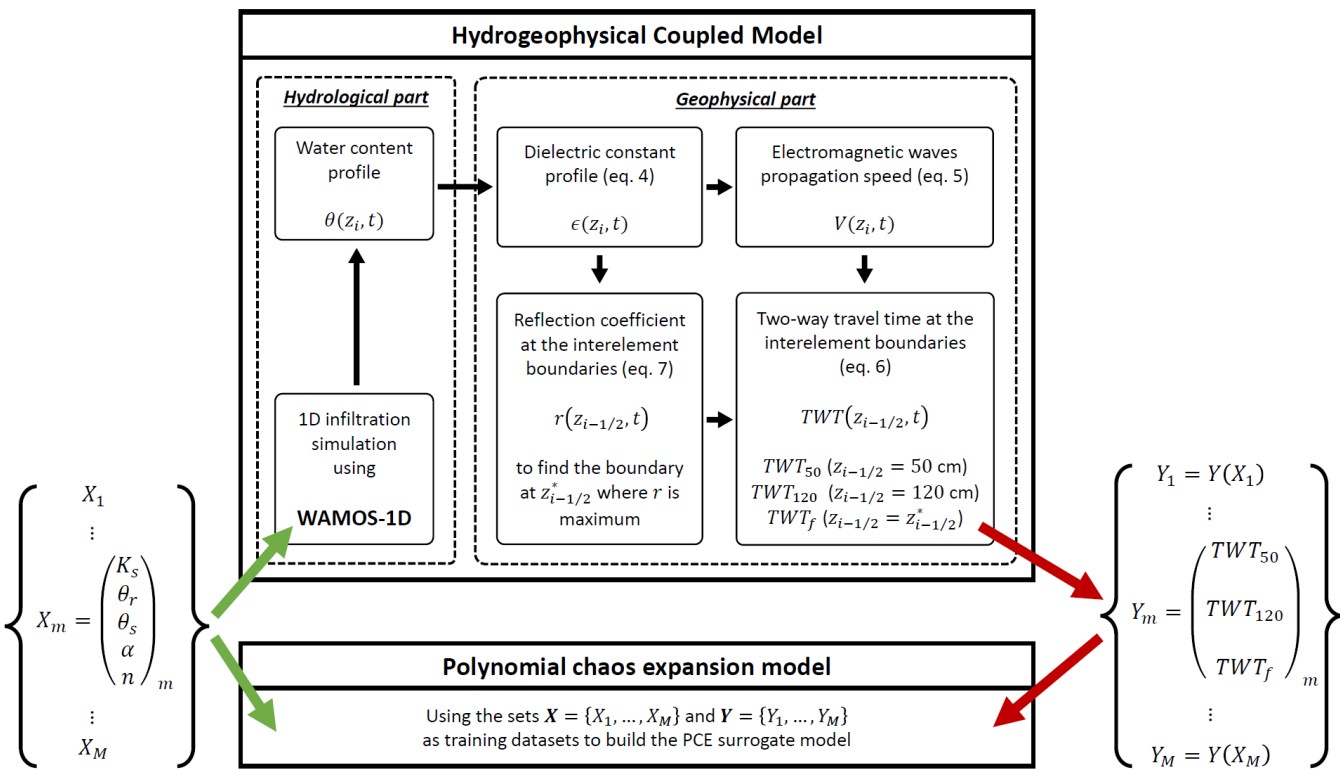

**Figure 3.** Summary of the working process of the forward hydrogeophysical model and how it is used to build the PCE surrogate model.

| | $K_s$ (cm/s) | $\theta_s$ (cm$^3$/cm$^3$) | $\theta_r$ (cm$^3$/cm$^3$) | $\alpha$ (1/cm) | $n$ (-) |
|---|---|---|---|---|---|
| $[x_{\min} - x_{\max}]$ | [0.001 - 0.15] | [0.32 - 0.48] | [0.01 - 0.13] | [0.01 - 0.28] | [1.5 - 10] |

**Table 1.** Prior intervals of the unsaturated soil parameters for both GSA and Bayesian estimation.

where $x_{-i} = \boldsymbol{X} \setminus x_i$ is the set of all parameters except $x_i$, $E()$ and $E(.|.)$ are the expectation and the conditional expectation operators, respectively, Var() and Var(.|.) are the variance and the conditional variance, respectively. The first-order index $S_i$ quantifies the contribution of the parameter $x_i$ alone to the total variance of $y(\boldsymbol{X})$, while $ST_i$ also includes all interactions of $x_i$ with the other parameters $x_{-i}$.

To perform a variance-based GSA, a practical approach (to save computational time) is to use Polynomial Chaos Expansion (PCE; Wiener, 1938). The PCE approach consists in developing any signal $y(\boldsymbol{X})$ as a set of orthonormal multivariate polynomials of a degree not exceeding $D$:

$$y(\boldsymbol{X}) = \sum_{|\beta| \leq D} s_\beta \Psi_\beta(\boldsymbol{X}) \tag{10}$$

|  | TWT$_f$ | TWT$_{50}$ | TWT$_{120}$ |
|---|---|---|---|
| **$t = 50$ s** | | | |
| Var_HYD_model | 14 | 5 | 15.9 |
| Var_PCE_model | 13.4 | 5 | 15.8 |
| Var_error | 4.3% | 0.9% | 0.5% |
| **$t = 150$ s** | | | |
| Var_hyd_model | 54.6 | 5.5 | 23.3 |
| Var_PCE_model | 53.9 | 5.4 | 23.2 |
| Var_error | 1.3% | 1.4% | 0.4% |
| **$t = 2000$ s** | | | |
| Var_hyd_model | 28.8 | 1.5 | 9.9 |
| Var_PCE_model | 27.1 | 1.4 | 9.3 |
| Var_error | 5.7% | 7.4% | 5.2% |

**Table 2.** Variance of TWT$_f$, TWT$_{50}$ and TWT$_{120}$ signals at $t = 50$ s, 150 s and 2000 s calculated with the PCE surrogate model and with the hydrogeophysical model.

where $\beta = \beta_1, \beta_2, ..., \beta_p \in \mathbb{R}^p$ is a $p^{\text{th}}$–dimensional index, $s_\beta$ are the PC coefficients, $\Psi_\beta$ are the generalized polynomial chaos of degree $|\beta| = \sum_{i=1}^{p} \beta_i$.

In this work, Legendre polynomials are used since uniform distributions are assumed for all uncertain parameters. Uniform distributions express the absence of prior information. This makes all parameter values in the given prior intervals equally likely. Large prior distribution intervals are considered for all unsaturated soil parameters (Table 1). Such combination of parameters investigated in the GSA is exhaustive and allows to consider a large panel of soil types. Notice that, while simulations with values of $n$ comprised between 1 and 1.5 would allow to investigate a wider range of porous media, they also take much longer to end.

The number of coefficients for a full PCE representation is $P = (p + D)!/p!D!$. A training dataset of $M$ realizations of the forward coupled hydrogeophysical model is used to build the PCE surrogate model of order $D$ (Fajraoui et al, 2011; Shao et al., 2017; Younes et al., 2013). The coefficients of the PCE are obtained by searching the best fit (in the least square sense) of the PCE surrogate model to the hydrogeophysical model for the $M$ realizations. To work with low-discrepancy sets, the $M$ realizations correspond to sets of input parameters sampled from their prior probability distributions, using quasi-random Sobol sequences (Shao et al., 2017). Because each parameter varies in its own range and has a proper unit, the parameter prior intervals are normalized to $[-1, 1]$ during PCE computation. We illustrate the principle of the construction of the PCE with our hydrogeophysical model in Fig.3.

A PCE is constructed at each time step for all model responses ($\text{TWT}_f$, $\text{TWT}_{50}$, and $\text{TWT}_{120}$) since we deal with transient simulations. In this work, $M = 2048$ hydrogeophysical model realizations are employed to obtain PCEs of degrees $D = 5$ containing $P = 252$ coefficients. The obtained PCEs are sufficiently accurate as the variance of the TWT output signals is calculated with the surrogate PCE model and the forward hydrogeophysical model at three different times $t = 50$ s, $150$ s, and $2000$ s. The results of Table 2 show that the relative difference between the two variances is very small for all investigated times. Note that although the relative variance error for the $\text{TWT}_{50}$ at $t = 2000$ s is the largest (around 7%), it remains insignificant since the total variance of the signal at this time is negligible (less than 2 $\text{ns}^2$). The variance of the forward hydrogeophysical model is therefore well reproduced by the PCE surrogate model which will be employed for the GSA of the TWT signals using the variance decomposition.

## 3.2 GSA results

The temporal distribution of the output variance of the three TWT signals ($\text{TWT}_f$, $\text{TWT}_{50}$ and $\text{TWT}_{120}$) are represented Fig.4. For each TWT signal, the variance is represented by the black curve and the relative contributions of the uncertain parameters to the variance are represented by the shaded area. The blank region between the variance curves and the shaded area represents interactions between parameters.

$\text{TWT}_f$ has a different behavior from the TWT signals of fixed reflectors $\text{TWT}_{50}$ and $\text{TWT}_{120}$ (Fig.4). Although the TWT signals of fixed reflectors have different variance magnitudes, they exhibit similar behavior (Fig.4b and 4c). The variance of the TWT signal is five times more significant for $\text{TWT}_{120}$ than for $\text{TWT}_{50}$. This is in agreement with the physics since the zone of the porous medium affecting the GPR wave is more important for the $\text{TWT}_{120}$ signal than for the $\text{TWT}_{50}$. In addition, the period of influence of the unsaturated parameters ($\theta_r$, $\alpha$, $n$) is also more important for $\text{TWT}_{120}$ than for $\text{TWT}_{50}$ since saturated conditions for the reflector R120 are reached much later than for R50. Since fixed reflectors exhibit similar behavior, in the following, we comment on the results of $\text{TWT}_f$ and $\text{TWT}_{120}$ signals.

### 3.2.1 GSA of the $\text{TWT}_f$ signal

$\text{TWT}_f$ variance is zero at the beginning of the infiltration (Fig.4a) which means that the $\text{TWT}_f$ signal is not affected by the initial conditions. Indeed, the infiltration wetting front and the $\text{TWT}_f$ signal start at zero for all parameter sets. Then, the variance of the signal increases until a maximum of 60 $\text{ns}^2$, reached at around 3 min. After that, the variance decreases, but keeps a significant value of around 25 $\text{ns}^2$ (Fig.4a). Concerning parameter sensitivities, at the beginning, the $\text{TWT}_f$ signal is mainly affected by $K_s$. The influence of this parameter decreases over time and reaches zero for long times when steady-state conditions (corresponding to a fully saturated soil) are reached. The parameter $\theta_s$ has a moderate influence on the $\text{TWT}_f$ signal. Its influence is not observable at short times since unsaturated conditions occur. Overall, the most influential parameter on the $\text{TWT}_f$ signal is the van Genuchten parameter $\alpha$. This parameter seems influential even for saturated conditions. Note that this numerical artifact is observed because the value of $\text{TWT}_f$ is artificially maintained when the infiltration wetting front reaches the water table, while physically the signal disappears. The effects of the parameters $\theta_r$ and $n$ are not observable (Fig.4a). The blank region between the variance curve and the shaded area in this figure is due to the interaction between the parameters.

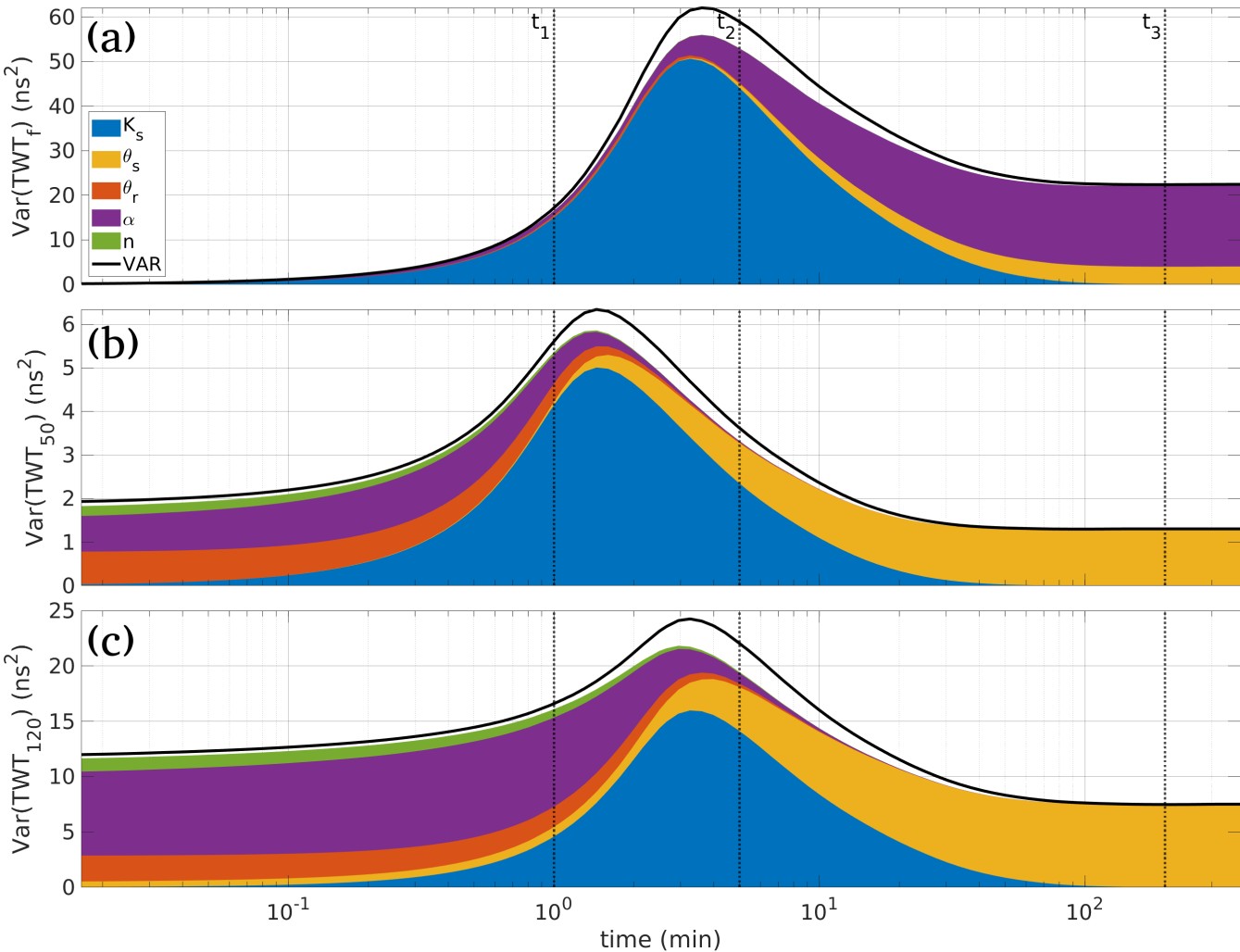

**Figure 4.** Time distribution of the variance of $TWT_f$ (a), $TWT_{50}$ (b) and $TWT_{120}$ (c). The shaded area under the variance curve represents the partial marginal contributions of the uncertain parameters; the blank region between the shaded area and the variance curves represents the contribution of interactions between the parameters. The marginal effects shown in Fig.6 are represented at three time steps $t_1 = 1$ min, $t_2 = 5$ min, and $t_3 = 200$ min, highlighted here (dotted black lines).

To estimate this interaction, we plot the difference between the total ($ST_i$) and the first order ($S_i$) sensitivity indices for all parameters (Fig.5a). This difference reflects the interaction between the parameters over time. Interactions between parameters are negligible for all parameters ($ST_i \approx S_i$), except for $K_s$ and $\alpha$ (Fig.5a). Hence, the interaction between these two parameters affects the variance of the $TWT_f$ signal as represented by the blank region between the variance curve and the shaded area (Fig.4a).

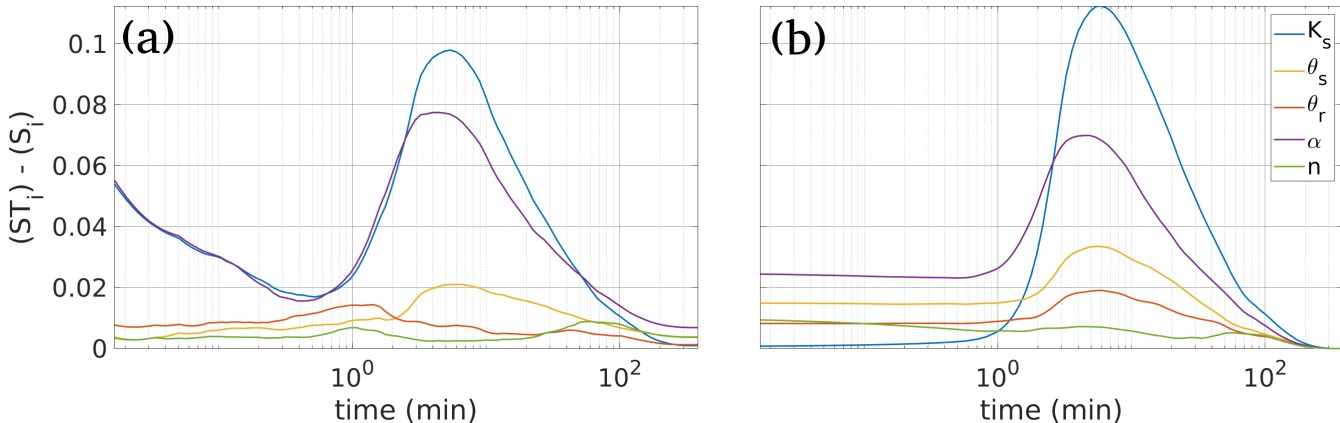

**Figure 5.** Difference between the total $(ST_i)$ and the first order $(S_i)$ sensitivity indices for all parameters for the $\mathrm{TWT_f}$ (a) and the $\mathrm{TWT_{120}}$ (b) signals.

To evaluate further the effect of the unsaturated soil parameters on the $\mathrm{TWT_f}$, we plot the marginal effect of each parameter (Fig.6). The marginal effect can be easily derived from the PCE coefficients and reflects the effect of one parameter on the output signal. Fig.6 depicts the marginal effects of each hydraulic parameter, i.e., their influence on the TWT signals as a function of their value when they vary over the range of their prior distribution interval, while the other hydraulic parameters are kept fixed at their center value. This representation allows determining the regions of influence of the hydraulic parameters, given that the stronger the slope of the marginal effect curve, the higher the influence of the parameter. These marginal effects can vary over time, so we represent them at the three time steps ($t_1 = 1$ min, $t_2 = 5$ min, and $t_3 = 200$ min) highlighted with dotted vertical black lines in Fig.4. The oscillations are caused by numerical artifacts related to the degree of the polynomials used in the PCE model. From Fig.6a, it can be noticed that:

- $K_s$ is highly influential at the beginning of the experiment. At $t_1 = 1$ min, the $\mathrm{TWT_f}$ signal varies almost linearly with $K_s$. Indeed, at the beginning of the experiment, when $K_s$ increases, the wetting front is more advanced, thus, the GPR wave propagates at a lower speed and the $\mathrm{TWT_f}$ signal increases. At $t_2 = 5$ min, the $\mathrm{TWT_f}$ signal is sensitive only for small $K_s$ values. Indeed, for high $K_s$ values, the soil is fully saturated and the perturbation of the high value of $K_s$ doesn't change the $\mathrm{TWT_f}$ signal. At $t_3 = 200$ min, the soil is fully saturated for almost all $K_s$ values, thus, the $\mathrm{TWT_f}$ signal becomes insensitive to $K_s$.

- $\theta_s$ has no influence at the first times ($t_1 = 1$ min) since unsaturated conditions occur. For long times, the $\mathrm{TWT_f}$ signal is very sensitive to $\theta_s$ with an almost linear behavior. Indeed, when the soil is fully saturated, the dielectric permittivity and thus the $\mathrm{TWT_f}$ signal is almost proportional to $\theta_s$.

- the sensitivity of $\mathrm{TWT_f}$ to $\theta_r$ is moderate and can be observed only at the beginning of the experiment (unsaturated conditions) with an almost linear behavior observable at $t = 1$ min and 5 min. The positive slope of the curve is consistent

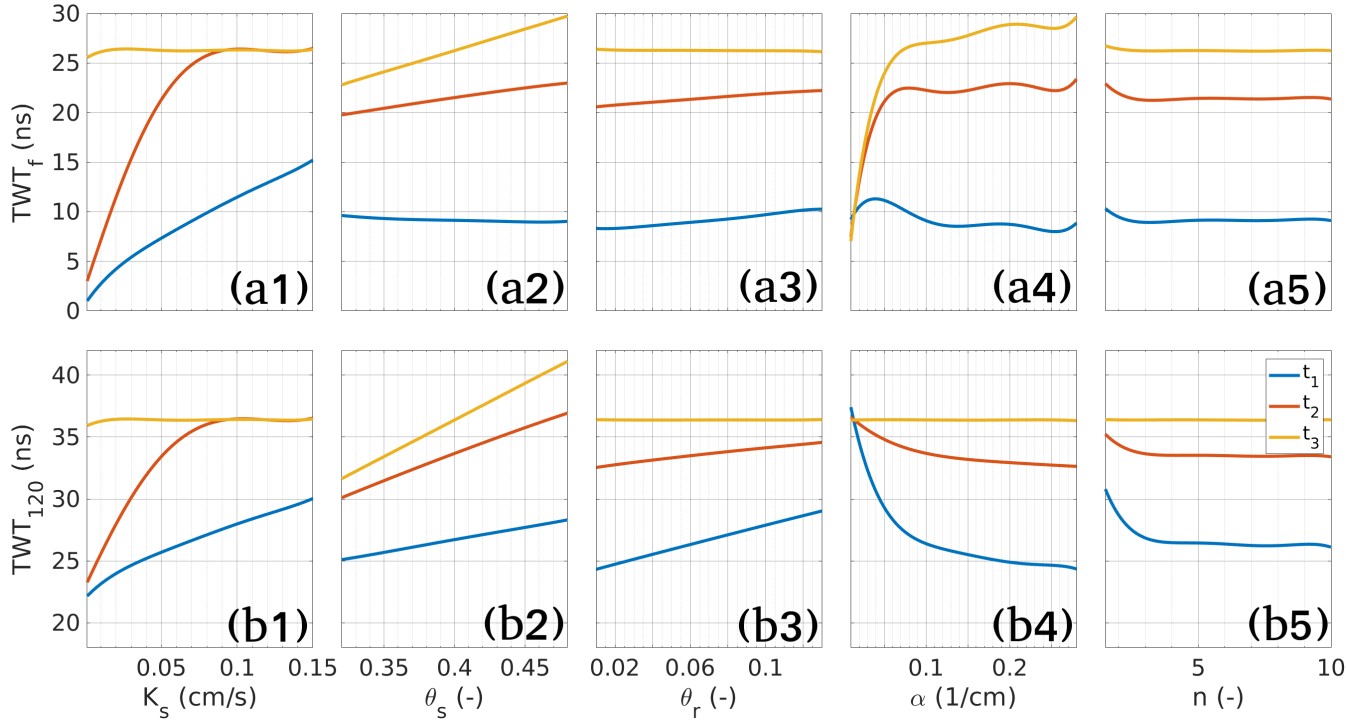

**Figure 6.** Marginal effects of the unsaturated soil parameters $K_s$, $\theta_s$, $\theta_r$, $\alpha$ and $n$ on the $\text{TWT}_f$ and $\text{TWT}_{120}$ signals at three different times $t_1 = 1$ min, $t_2 = 5$ min, and $t_3 = 200$ min, highlighted in Fig.4.

with the physics of the process (when $\theta_r$ increases, the speed of the electromagnetic wave decreases, and the $\text{TWT}_f$ signal increases).

– The van Genuchten parameter $\alpha$ is highly influential notably for long times ($t_3 = 200$ min). A small variation of the parameter $\alpha$ can induce a strong variation of the $\text{TWT}_f$ signal. Notably, the sensitivity of $\alpha$ is very high for $\alpha \leq 0.05$ cm$^{-1}$.

– The sensitivity of $\text{TWT}_f$ to the parameter $n$ is almost zero (flat curves) at all times ($t = 1$ min, 5 min, and 200 min). The parameter $n$ has therefore a negligible effect on the $\text{TWT}_f$ signal and, as a consequence, it is expected to be poorly identifiable from the $\text{TWT}_f$ data. $\text{TWT}_f$ is sensitive to all $K_s$, $\theta_s$ and $\theta_r$ values tested (see table 1), but is also sensitive to alpha < 0.05 cm$^{-1}$ and n < 2. This means a wide range of soil types.

### 3.2.2 GSA of the $\text{TWT}_{120}$ signal

The variance of the $\text{TWT}_{120}$ signal is nonzero at the beginning of the experiment which means that the $\text{TWT}_{120}$ signal is affected by the initial conditions (Fig.4c). Indeed, at the very beginning, the pressure distribution is hydrostatic and the water content distribution in the column is obtained from Eq.2 which depends on all soil parameters except $K_s$. Therefore, the speed

of the GPR wave depends on the initial water content distribution which is dependent on the unsaturated soil parameters $\theta_s$, $\theta_r$, $\alpha$, and $n$. The most influential parameter at the beginning of the experiment is the parameter $\alpha$. Over time, the effect of this parameter reduces, whereas the effect of $\theta_s$ increases. For long times, $\theta_s$ becomes the only sensitive parameter. The parameter $K_s$ is also very sensitive. Its effect starts at zero, and increases until a maximum is reached at around 3 min, then it slowly decreases and becomes negligible after 100 min. As with the TWT$_f$ signal, interactions between parameters are moderate. The

difference between the total and first-order Sobol indices is negligible for all parameters except after 1 min for the parameters $K_s$, $\alpha$ and $\theta_s$ (see Fig.5b). This interaction corresponds to the blank region, between the variance curve and the shaded area in Fig.4c. The marginal effects of the soil parameters on the TWT$_{120}$ signal are plotted in Fig.6b for $t = 1$ min, 5 min, and 200 min. The curves in this figure show that:

– As for the TWT$_f$ signal, $K_s$ is highly sensitive, especially for $t = 1$ min and 5 min.

– The saturated water content $\theta_s$ is very influential for all times. The TWT$_{120}$ varies almost linearly with $\theta_s$ even at the beginning ($t_1 = 1$ min), since the fixed reflector is located in the lower saturated region.

– As for the TWT$_f$ signal, $\theta_r$ is sensitive only at the beginning of the experiment (unsaturated conditions) with an almost linear behavior at $t = 1$ min and 5 min. When $\theta_r$ increases, the water content increases, and hence, the TWT$_{120}$ increases.

– The van Genuchten parameter $\alpha$ is highly sensitive. However, contrarily to the TWT$_f$ signal where $\alpha$ is highly sensitive

at long times ($t_3 = 200$ min), the sensitivity of $\alpha$ for the TWT$_{120}$ signal is high at short times ($t_1 = 1$ min). For long times, the influence of $\alpha$ disappears since the soil becomes fully saturated. The negative slope of the curve of the TWT$_{120}$ signal as a function of $\alpha$ observed at the beginning of the experiment is consistent with the physics of the process. Indeed, when $\alpha$ increases, the capillary fringe thickness decreases, hence, the water content in the unsaturated zone decreases, and thus the TWT$_{120}$ signal decreases.

– Surprisingly, and contrarily to the TWT$_f$ signal which showed a flat curve for the marginal effect of the parameter $n$ for all parameter values and at all investigated times, the TWT$_{120}$ signal is sensitive to $n$ at the beginning of the experiment ($t_1 = 1$ min) with a high sensitivity for $n < 3.5$ and a moderate sensitivity (the curve has a small slope) for $n \geq 3.5$. Finally, TWT$_{120}$ shows similar sensitivity for $K_s$, $\theta_s$ and $\theta_r$ but slightly higher than TWT$_f$. For $\alpha$, they show complementarity which makes the procedure very efficient for $\alpha < 0.05$ using all the infiltration experiment (early time for TWT$_f$ and late

time for TWT$_{120}$). TWT$_{120}$ is more sensitive to $n$, but until values 3.5. The GSA study shows that the monitoring of the infiltration using both the TWT from the infiltration front and (at least) a fixed reflector shows a significant sensitivity for a wide range of soil types (see in Table 1 the hydraulic parameters range tested). Next section demonstrates the use of this sensitivty to calibrate these parameters.

## 4 Bayesian soil parameter estimation from the TWT signals

In this section, we estimate the unsaturated soil parameters in a Bayesian framework using the Markov Chain Monte Carlo (MCMC) sampler (Vrugt and Bouten, 2002; Vrugt et al., 2008). The statistical calibration is performed for a GPR monitored infiltration experiment in order to address the following questions:

1. Can we obtain an appropriate estimation of all unsaturated soil parameters from TWT data?

2. What is the impact of the kind of TWT data (moving/fixed reflectors) and of the number of reflectors on the calibrated model parameters and their confidence intervals?

3. What is the optimal set of TWT measurements that yields good reliability of all unsaturated soil parameters?

The MCMC method has been successfully employed in various inverse hydrological problems (e.g., Fajraoui et al., 2011; Younes et al., 2016; Younes et al., 2017; Younes et al., 2018). The method generates random sequences of parameter sets that asymptotically converge toward the target joint posterior distribution by searching the ensemble of possible parameter sets that satisfactorily fit the observations. The converged sets can then be used to assess the quality of the parameter estimation such as the optimal parameter values and the 95% Confidence Intervals (CIs) which allow for evaluating the reliability of the parameters via uncertainty quantification.

In the sequel, the MCMC method is performed with the DREAM$_{(ZS)}$ (DiffeRential Evolution Adaptive Metropolis) software (Laloy and Vrugt, 2012; Vrugt, 2016). This software samples the posterior probability density function (pdf) by running multiple Markov chains simultaneously for global exploration of the parameter space. The prior distributions of the parameters are the same than in the GSA (Table 1). The DREAM$_{(ZS)}$ then automatically tunes the scale and orientation of the proposal distribution until we get the posterior target pdf. A MATLAB toolbox of the DREAM$_{(ZS)}$ algorithm is available for Bayesian inference in fields ranging from physics, chemistry and engineering, to ecology, hydrology, and geophysics. The vector of unknowns is formed by the five unsaturated soil parameters ($K_s$, $\theta_s$, $\theta_r$, $\alpha$, $n$). Compared to the GSA, which allows an investigation of a large panel of soil types, the parameter estimation is demonstrated on a single synthetic case. A reference solution is generated by simulating the hydrogeophysical problem formed by the system of equations (1)-(6) using the following reference parameter values $K_s^* = 0.08$ cm/s, $\theta_s^* = 0.4$, $\theta_r^* = 0.07$, $\alpha^* = 0.145$ cm$^{-1}$, $n^* = 2.68$, as shown in Table 3. These parameter values corresponds to those of a sandy porous medium present in an experimental platform where we test the protocol under real conditions. The modeled TWT$_f$, TWT$_{50}$, and TWT$_{120}$ signals used as synthetic calibration data are deduced from the results of the simulation using the reference parameter values. These TWT signals are then independently corrupted using a normally distributed noise with a standard deviation $\sigma = 0.5$ ns. This error corresponds to an uncertainty of 1 ns, which is realistic in the instance of an 800 MHz GPR antenna.

The TWT$_f$, TWT$_{50}$ and TWT$_{120}$ calibration signals, illustrated before noise corruption in Fig.2, increase almost linearly until reaching a plateau. For the TWT$_{50}$ signal, the plateau is reached when the infiltration front attains the R50 reflector and the value of the plateau corresponds to the time needed by the electromagnetic wave to make a round trip from the surface to a 50 cm depth of a full saturated porous medium. For the TWT$_{120}$ signal, the plateau signal is reached when the infiltration front

|  | $K_s$ (cm/s) | $\theta_s$ (-) | $\theta_r$ (-) | $\alpha$ (1/cm) | $n$ (-) |
|---|---|---|---|---|---|
| **$X^*$** | 0.08 | 0.40 | 0.07 | 0.145 | 2.68 |
| **Scenario 1** | **0.081** | **0.39** | **0.076** | **0.211** | **2.75** |
| $TWT_f$ | (0.037) | (0.031) | (0.14) | (0.167) | (0.93) |
|  | *4* | *5* | *1* | *2* | *9* |
| **Scenario 2** | **0.074** | **0.4** | **0.081** | **0.173** | **5.79** |
| $TWT_{50}$ | (0.023) | (0.008) | (0.061) | (0.269) | (9.99) |
|  | *6* | *19* | *2* | *1* | *1* |
| **Scenario 3** | **0.078** | **0.4** | **0.089** | **0.167** | **5.93** |
| $TWT_{120}$ | (0.011) | (0.007) | (0.053) | (0.195) | (9.36) |
|  | *13* | *24* | *2* | *1* | *1* |
| **Scenario 4** | **0.08** | **0.4** | **0.074** | **0.151** | **2.72** |
| $TWT_f$, | (0.003) | (0.004) | (0.015) | (0.029) | (0.5) |
| $TWT_{120}$ | *46* | *37* | *8* | *9* | *17* |
| **Scenario 5** | **0.079** | **0.4** | **0.073** | **0.149** | **2.68** |
| $TWT_f$, | (0.003) | (0.004) | (0.015) | (0.027) | (0.49) |
| $TWT_{50}$, | *49* | *44* | *8* | *10* | *17* |
| $TWT_{120}$ |  |  |  |  |  |

**Table 3.** First line: Reference values used to build the synthetic calibration data. Then for the different scenarios: estimated mean values (bold), size of the posterior confidence intervals (CIs) (between brackets), and ratio of prior to posterior intervals (italic).

attains the water table (the domain becomes fully saturated) and the value of the plateau corresponds to the time needed by the electromagnetic wave to make a round trip from the surface to a 120 cm deep of a fully saturated porous medium. For $TWT_f$, the plateau value is also reached when the infiltration front attains the water table and the value of the plateau corresponds to the time needed by the electromagnetic wave to make a round trip from the surface to the water table at 100 cm deep.

The reliability of the unsaturated soil parameters is assessed for 5 different scenarios of measurement sets. In the first scenario, only data of the wetting-front $TWT_f$ signal are used for the calibration. The second and third scenarios use only the $TWT_{50}$ and $TWT_{120}$ signal, respectively, obtained from reflection on the fixed reflector R50 and R120. The fourth scenario uses both data of $TWT_f$ and $TWT_{120}$ as fitting data. The last scenario investigates the benefit of adding a fixed reflector by using data of the $TWT_f$, $TWT_{50}$ and $TWT_{120}$ signals as conditioning information.

In the five scenarios, the MCMC sampler uses three parallel chains and a total number of 50000 runs. The last 25% of the runs that adequately fit the model onto observations are used to estimate the joint posterior distribution.

The MCMC results of the five studied scenarios are given in Table 3 which depicts, for each parameter, the mean estimated value, its posterior CI size, and the ratio of prior to posterior intervals. Note that the CI and the last indicator are calculated from the standard deviation by assuming a Gaussian posterior distribution. A small CI indicates an accurate estimation of the parameter. A significant difference between the prior and posterior intervals is a sign of the high sensitivity of the model to that parameter (Dusek et al., 2015). The posterior histograms and the derived statistics are obtained from the the last 12500 simulations, as mentionned above, for which the Gelman Rubin (Gelman and Rubin, 1992) criterion is verified and the chains are stable and not autocorrelated.

Results of table 3 for scenario 1 using only data of the $TWT_f$ signal for the estimation of the unsaturated soil parameters show that:

- An accurate estimation of $K_s$, the most sensitive parameter (Fig.4a), is obtained with a CI of 0.037 cm/s and a variation interval reduced by 4.

- A fair estimate of the parameters $\theta_s$ with a standard deviation of 0.031 (-) and a reduction of the interval of variation by 5. This result is relatively surprising as this parameter did not show a strong influence on $TWT_f$ sensitivity (Fig.4a).

- The parameter $\theta_r$ is not well estimated. Indeed, although its mean estimated value is very close to its reference value, the associated uncertainty of 0.14 is large and the posterior interval is as large as the prior one, which indicates the low reliability of the estimation.

- A poor estimation of $\alpha$, while the sensitivity analysis showed it has a strong influence on $TWT_f$ (Fig.4a). Its CI is large, with a value of 0.167 cm$^{-1}$ and its posterior interval size is half the prior one.

- The $TWT_f$ signal yields a mean estimated value $n = 2.75 \pm 0.47$ which is close to the reference value $n^* = 2.68$. The parameter $n$ is quite well identified since its posterior interval is 9 times smaller than the prior interval. This is relatively surprising since the sensitivity of $n$ is negligible (Fig.4a and 6a5). $n$ values comprised between 1.5 and 3 could represent silty loam, sandy loam or sand. Therefore, it is not obvious to consider this result as a good identification. We have performed another estimation (the results are not presented here) with a target value of the $n$ parameter equal to 6, which is located in the low sensitivity region of the parameter. In this case, the inversion led to a relatively good estimation (estimated mean value of 6.97). However, the parameter was poorly identified since its posterior interval was still large, though it was a bit smaller than the prior parameter interval.

In summary, using only data of the $TWT_f$ signal as conditioning information for the hydrogeophysical model calibration yielded well mean estimated parameter values, close to the reference values for all unsaturated soil parameters. However, the examination of the associated uncertainties, showed that only $K_s$, $\theta_s$, and $n$ are correctly identified (with narrow posterior intervals with respect to the prior ones). This points out the importance of statistical calibration methods for highly nonlinear problems to investigate not only estimated parameter values but also the associated uncertainties.

The estimation of the unsaturated soil parameters for scenarios 2 and 3, using only data of the $TWT_{50}$ or $TWT_{120}$ signal for the calibration shows that:

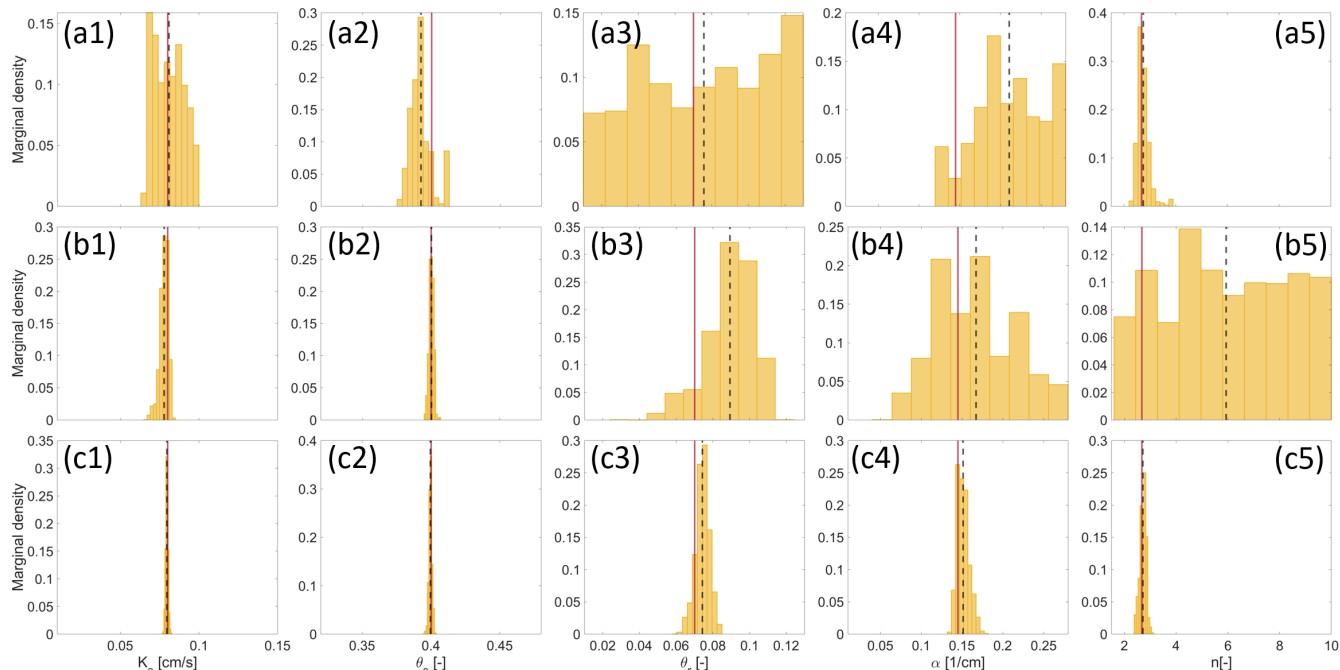

**Figure 7.** MCMC solutions using scenarios 1 (a1-a5), 3 (b1-b5), and 4 (c1-c5) for calibrating the hydrogeophysical model. The histograms are built from the posterior distributions. The estimated mean values are represented in dotted black line and compared to the exact target value (hard red line). The displayed parameter intervals correspond to the prior upper and lower limits of Table

1.

– The parameters $K_s$ and $\theta_s$, which are the most sensitive parameters during most of the experiment (Fig.4b and 4c), are well identified with small CI size and strong reductions by at least 6 for $K_s$, and 19 for $\theta_s$, of their intervals of variation. We note that the TWT$_{120}$ signal allows a much better estimate of both $K_s$ and $\theta_s$ as their CIs are smaller than the ones estimated with TWT$_{50}$. It is especially true for $K_s$ where there is almost a factor 2 between the reduction ratios.

– The soil parameters $\theta_r$, $\alpha$, and $n$, although sensitive (Fig.4b and 4c), cannot be identified from the TWT$_{50}$ and TWT$_{120}$ signals since their posterior intervals are as large as, or at best two times smaller than their prior ones.

The results of scenario 4 which combines data of TWT$_\text{f}$ and TWT$_{120}$ signals show that:

– Both parameters $K_s$, $\theta_s$, and $n$ are very well identified, with very narrow posterior intervals showing a strong reduction by 46, 37, and 17 of their prior intervals, respectively.

– The parameters $\theta_r$ and $\alpha$ are reasonably well estimated with mean values very close to their reference and intervals of variation reduced by 8 and 9, respectively.

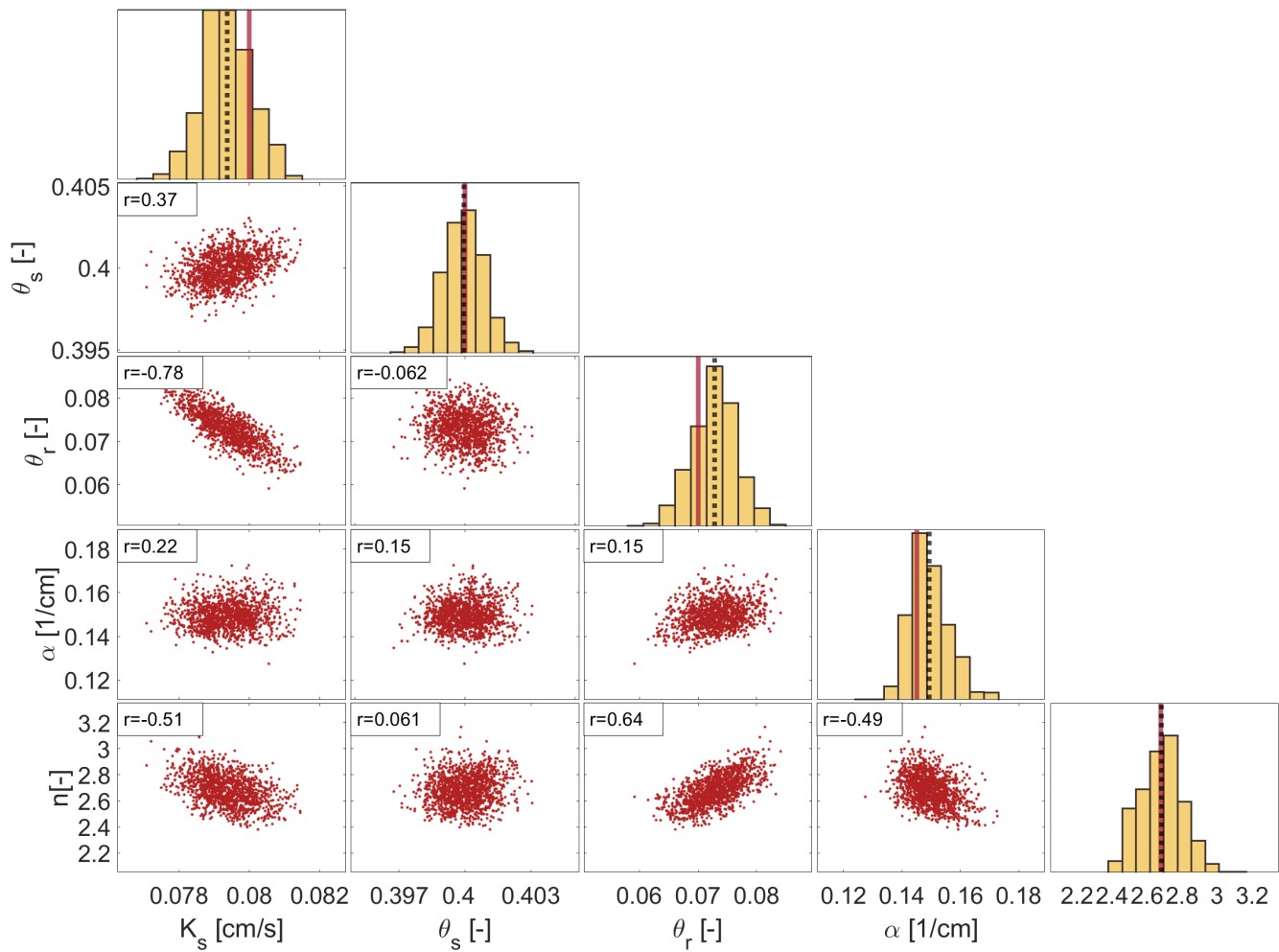

**Figure 8.** MCMC solutions using TWT$_f$, TWT$_{50}$ and TWT$_{120}$ signals for the calibration of the hydrogeophysical model. The diagonal plots represent the inferred posterior parameter distributions, showing the estimated mean value (dotted black line) and the target value (hard red line). The off-diagonal represents the pairwise correlations between parameters.

Fig.7 shows the posterior histograms obtained from the scenarios 1, 3, and 4. For all parameters, the displayed intervals correspond to the prior upper and lower limits of Table 1.

Finally, the results of the last scenario which combines data of TWT$_f$, TWT$_{50}$ and TWT$_{120}$ signals, show performances very similar to scenario 4. Additional information from TWT$_{50}$ helped to reduce slightly the posterior intervals of $K_s$, $\theta_s$, and $\alpha$ that in that case show a reduction of 49, 44, and 10 times their prior intervals, respectively.

The results of MCMC for this last scenario are shown in Fig.8 where diagonal plots depict the inferred posterior parameter distributions and the off-diagonal scatterplots represent the pairwise correlations in the MCMC draws. Almost bell-shaped

posterior distributions are obtained for all unsaturated soil parameters. Negligible correlations are observed between the parameters, except moderate correlations observed between $K_s$ and $\theta_r$ (r=-0.78) and between $n$ and $\theta_r$ (r=0.64) .

Note that the parameter $n$ is relatively well estimated as the target reference value 2.68 is located in the high sensitivity region ($n < 3.5$) (Fig.6). In the case of a reference value located in the low sensitivity region ($n \geq 3.5$), the calibration of the hydrogeophysical model using $TWT_f$ and $TWT_{120}$ signals yields a much poorer identification of the parameter $n$. For instance, using scenario 5 with a reference value $n^* = 4.25$, the estimated mean value is 4.84 with a posterior CI size of 3.6, which corresponds to a reduction of the interval of variation by only 2.

To complete the numerical study, the protocol was tested varying the boundary conditions. One can wonder how much the thickness of the vadose zone would impact the calibration of the hydraulic parameters. For that purpose, 3 scenarios have been considered, varying the water table depth from 50 cm to 1 m and 2 m and assuming a hydrostatic initial profil. Results of the MCMC calibration depicted in Fig.9 show that the 5 parameters are even better estimated when the water table is deeper. We explained this result because when the vadose zone is thicker, then the initial water content profile highlights a larger variation with depth which is perturbed when the infiltration propagates. In the shallowest case, with a 50 cm deep water table, $\alpha$ and $n$ could not be recovered because the water content (which directly affects the radar propagation) in the vadose zone is already close to the saturation conditions. One should note that in this case, we maintained the fixed reflector at 120 cm depth, which means it is above the water table in the 2 m case. In this latter case, a deeper fix reflector would enhance the result (as seen in previous scenarii) but in field conditions, a deeper fixed reflector would be harder to be reliably detected. Nevertheless, we show here that it is not necessary to have a reflector below the water table to obtain an accurate calibration. We also vary the surface boundary conditions by using different heights of water ponding at the surface, which would pratically mean varying the height of the infiltration ring, from 5 cm to 10 cm and 20 cm. As one could expect, it only affects the duration of the infiltration experiment without impact on the accuracy of the hydraulic parameters calibration. The infiltration duration could be divided by 2 if the pressure head is doubled. This is worth to notice, especially for medium with low permeability, to speed-up the experiment.

Last, the efficiency of the protocol was numerically tested on 3 types of soils used in the experimental platform SCERES in Strasbourg (Bohy et al., 2006). A coarse, a medium and a fine sand are considered (see Table 4). The permeability varies over 2 orders of magnitude, $\theta_r$ is multipied by 4, $\alpha$ if multiply by 10 and n varies from 2.0 to 2.7. The results of the calibration using the $TWT_f$ and $TWT_120$ summarized in Table 4 shows that the parameters for the 3 materials are well estimated, even better when the sand is finer. All 5 hydraulic parameters are recovered here, considering a water table at 1 m depth.

These results evidences that the GPR signal data of both the wetting front and a fixed reflector can bring very different but complementary information for the identification of the unsaturated soil parameters. They also point to the high benefit of combining the GPR signal data of a fixed reflector, preferably located sufficiently deep in the soil, with the TWT signal of the moving infiltration wetting front. This combination allows good reliability of almost all soil parameters with very narrow posterior intervals in comparison with the prior ones. In particular, the van Genuchten parameter $n$ is relatively well identified for investigated sandy soil where $n < 3.5$.

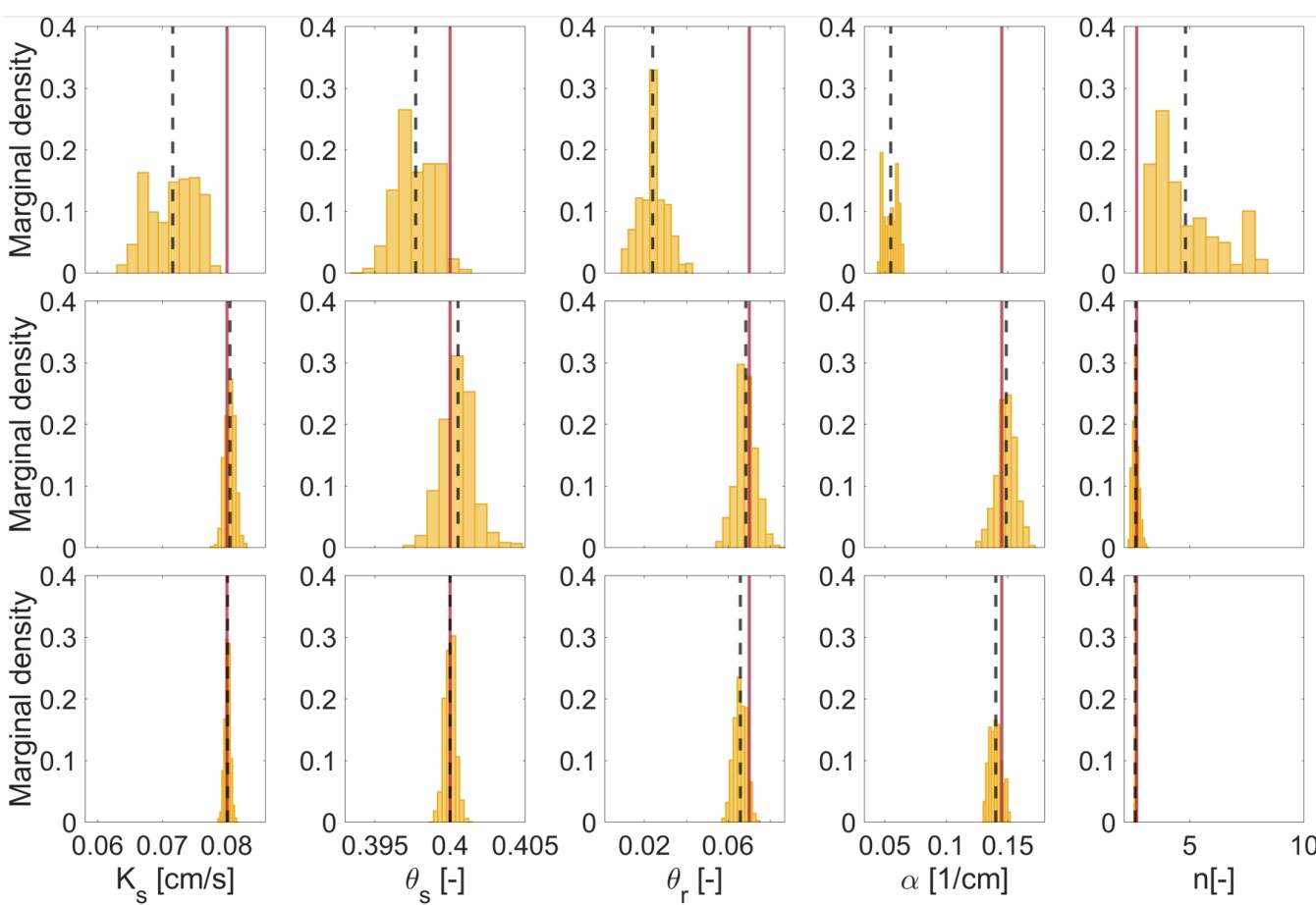

**Figure 9.** MCMC solutions considering a water table at 50 cm, 1m and 2m depth (from top to bottom) for calibrating the hydrogeophysical model. The histograms are built from the posterior distributions. The estimated mean values are represented in dotted black line and compared to the exact target value (hard red line). The displayed parameter intervals are equal from each parameters (each column).

## 5 Conclusions

The aim of the present study was to optimize a cheap method used at the field scale to characterize the hydraulic parameters of the porous medium. To this end, we investigated a particular protocol: time-lapse GPR monitoring of artificial infiltration experiments. Water infiltration into an initially unsaturated sandy soil has been simulated using a one-dimensional hydrogeophysical model. GPR time signals have been analyzed from the reflection of the electromagnetic wave on the moving wetting front and on two fixed reflectors located at different depths. GSA, based on PCE decomposition, has been used to assess the effect of the unsaturated soil parameters (saturated hydraulic conductivity, saturated and residual water contents, and Mualem–van Genuchten shape parameters $\alpha$ and $n$) on the different TWT signals. Statistical calibration of the unsaturated soil parame-

|  | $K_s$ (cm/s) | $\theta_s$ (-) | $\theta_r$ (-) | $\alpha$ (1/cm) | $n$ (-) |
|---|---|---|---|---|---|
| | | | **Coarse sand** | | |
| $\boldsymbol{X^*}$ | 0.6 | 0.38 | 0.15 | 0.145 | 2.0 |
| | **0.578** | **0.379** | **0.161** | **0.21** | **1.947** |
| | (0.109) | (0.013) | (0.081) | (0.194) | (1.205) |
| | *9* | *12* | *2.5* | *1.5* | *7* |
| | | | **Medium sand** | | |
| $\boldsymbol{X^*}$ | 0.08 | 0.40 | 0.07 | 0.145 | 2.68 |
| | **0.08** | **0.399** | **0.069** | **0.147** | **2.59** |
| | (0.003) | (0.003) | (0.02) | (0.027) | (0.426) |
| | *47* | *53* | *10* | *10* | *20* |
| | | | **Fine sand** | | |
| $\boldsymbol{X^*}$ | 0.005 | 0.43 | 0.036 | 0.016 | 2.5 |
| | **0.005** | **0.43** | **0.049** | **0.016** | **2.511** |
| | (0.0) | (0.002) | (0.05) | (0.001) | (0.465) |
| | *300* | *80* | *4* | *279.5* | *18.5* |

**Table 4.** Hydraulic parameters for 3 types of sand, a coarse, medium and fine sand (from top to bottom): target values, the estimated mean values, size of the posterior confidence intervals (CIs) and ratio of prior to posterior intervals.

ters has been performed with the MCMC sampler using corrupted synthetic observations to evaluate the reliability of the soil parameters from the TWT signals.

The results of GSA showed that the TWT$_f$ signal of the wetting front is different from that of the two fixed reflectors which had similar behavior. For the fixed reflectors, the magnitude of the variance (and therefore the sensitivity of the soil parameters) is more pronounced for deeper reflectors. The TWT$_f$ signal is highly sensitive to $K_s$ and $\alpha$ and moderately sensitive to $\theta_s$. A low sensitivity was observed for $\theta_r$, whereas the parameter $n$ was insensitive. The TWT$_{120}$ signal of the fixed reflector located at 120 cm depth is highly sensitive to $K_s$, $\theta_s$ and $\alpha$, and moderately sensitive to $\theta_r$. The van Genuchten parameter $n$ has a high sensitivity for $n < 3.5$ and a poor sensitivity for $n \geq 3.5$. The GSA study shows that the monitoring of the infiltration using both the TWT from the infiltration front and (at least) a fixed reflector has a significant sensitivity for a wide range of soil types.

The reliability of the unsaturated soil parameters has been assessed for 5 different scenarios of TWT measurement sets. When only data of the TWT$_f$ signal are used as conditioning information for the model calibration, all estimated parameter values were very close to the reference values. However, analyzing the associated uncertainties showed that only $K_s$, $\theta_s$, and $n$ were correctly identified (with narrow posterior intervals). Further, using only data of the TWT$_{50}$ or TWT$_{120}$ signals for the calibration allows also only a good identification of $K_s$ and $\theta_s$ with a strong reduction of their intervals of variation. The best results, in terms of parameter reliability, are obtained with the combination of TWT$_f$ with at least one fixed reflector. In this

case, the four parameters $K_s$, $\theta_s$, $\theta_r$, and $\alpha$ are very well identified with very narrow posterior intervals. The van Genuchten parameter $n$ is estimated with a low uncertainty but its estimation degrades in the low sensitivity region $n \geq 3.5$. We note that the deeper reflectors provide more information as the inversion of its signal furnishes parameters with lower uncertainty. Then using two or three reflectors in addition to the wetting front signal doesn't reduce consequently the uncertainty of the parameters. The procedure has been applied for 3 types of soil ranging from coarse to fine sand and the results of MCMC simulations have shown its efficiency. The best estimate was obtained for the finest material. In field condition, one could expect a higher clay content, which would decrease the electrical resistitivy and then would attenuate the GPR signal, limiting the penetration depth of the radar wave. Hence, our numerical study shows that using a higher infiltration ring and applying a greater pressure head could speed-up the protocol without any impact on the MCMC results. Furthermore it appears that a deeper water table makes the calibration protocol more efficient and accurate. A limitation is observed for very shallow water table (for instance 50 cm) where the van Genuchten parameters $\alpha$ and $n$ could not be estimated because the vadose is already almost saturated.

The results of this study highlight the high benefit of combining TWT signals of fixed and moving (infiltration wetting front) reflectors for very good identification of all the unsaturated soil parameters. It also points out the role of GSA to assess the influence of the parameters on the output signals and the necessity to perform statistical calibration to assess the reliability of model parameters by evaluating not only estimated parameter values but also their associated uncertainties.

### Acknowledgements

The doctoral position of the first author is co-funded by the Grand-Est Region and the University of Strasbourg. This research work is partly funded by the French National Research Agency through the Exciting project ANR-17-CE06-0012-01. Finally, computing time was provided by the HPC-UdS.

*Code availability.* The hydrological code WAMOS-1D, the coupled hydrogeophysical model, and the global sensitivity analysis tools can be provided upon request by the authors.

*Data availability.* All data presented in this paper is available upon request from the authors.

*Author contributions.* JFG and NL formulated the aims of the research. BB, FL, and AY provided the hydrological code. AY provided the resources for sensitivity analysis and Bayesian estimation studies. RM established the hydrogeophysical coupled model and ran all experiments under the supervision of JFG and NL, and validation of all co-authors. RM prepared the manuscript with reviewing and editing contributions from all co-authors.

*Competing interests.* The authors declare that they have no conflict of interest.

*Acknowledgements.* The doctoral position of the first author is co-funded by the Grand-Est Region and the University of Strasbourg. This research work is partly funded by the French National Research Agency through the Exciting project ANR-17-CE06-0012-01. The authors would like to acknowledge the High-Performance Computing Center of the University of Strasbourg for supporting this work by providing scientific support and access to computing resources. Part of the computing resources was funded by the Equipex Equip@Meso project (Programme Investissements d'Avenir) and the CPER Alsacalcul/Big Data.

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
