# Peer review of "Coupled hydrogeophysical inversion of an artificial infiltration experiment monitored with ground penetrating radar: synthetic demonstration"

_EGUsphere, 2022_

## Author Comment (AC2)

**Final Authors' Comments**

**Title:** On the interest of ground penetrating radar data for the estimation of unsaturated soil parameters
**Author(s):** Rohianuu Moua et al.
**MS No.:** egusphere-2022-936
**MS type:** Research article

Reviewers' comments are written in black.

*Responses to reviewers' comments are written in italic blue.*

*Reviewers' comments that are shaded (in grey) are suggested corrections that will be applied in the corrected version of the manuscript.*
* * *
**Reviewer 1**

**General comments**

The authors analyze, through a numerical experiment, the retrieval of the unsaturated soil hydraulic properties from ground-penetrating radar (GPR) travel times corresponding to the wetting front as well as to fixed reflectors during an infiltration event. The hydrodynamic simulation involves a 1D solution of Richard's equation and the top and bottom boundary conditions are a 10 cm pressure head (Dirichlet) and a 1 m deep water table, respectively. Only one soil type, sand-like, is analyzed. The radar system reduces to a point at the soil surface, under water, and propagation times are calculated from propagation velocities derived from water content. The radar-antenna system and interactions with the medium are not accounted for. Sensitivity analyses as well as parameter estimation using the Markov Chain Monte Carlo Bayesian approach show that travel time information indeed provides valuable information to estimate the soil hydraulic properties. Different parameter sensitivities and corresponding uncertainties are observed and discussed.

The paper is well written and presented. It is technically sound and of interest to the scientific community. Nevertheless, compared to the state of the art, its novelty is quite limited. It is a case study and the interest mainly lies in the specific boundary conditions that are used for the hydrodynamic event as well as to the corresponding analyses. The fact that the radar and radar-medium interactions are not modeled limits the scope of the conclusions for real applications. The physical interpretation of the results could be deepened. The state of the art close to the topic of interest should be reviewed and links should be made with the observations of the authors.

*We thank the referee for his very constructive comments. In a new version we will enlarge the state of the art, deepen the physical interpretation of the results and relate them to the state of the art.*

**Specific comments**

Title and state of the art: The title should be more informative regarding the content of the paper. Indeed, the interest of ground penetrating radar to identify the soil hydraulic properties has already been demonstrated by many studies during these last two last decades. In that respect, a deeper literature review should be made to demonstrate the interest of the present study. Several studies relatively close to the topic of this paper are not referred to.

*The title will be changed to:"Coupled Hydrogeophysical inversion of an artificial infiltration experiment monitoring by ground penetrating radar: synthetic demonstration."*

*Some references, already cited in the manuscript but not compared to the present study, will be further discussed in the introduction section to emphasize the novelty of this research, such as: Saintenoy et al., 2008; Moysey, 2010; Scholer et al., 2011; Jadoon et al., 2012; Busch et al., 2013; Leger et al., 2014; Jonard et al., 2015; Leger et al., 2016; Leger et al., 2020*
*New references will also be added, with for instance:*
*Jaumann, S. and K. Roth (2018). "Soil hydraulic material properties and layered architecture from time-lapse GPR". In: Hydrology and Earth System Sciences 22.4, pp. 2551–2573. DOI: 10.5194/hess-22-2551-2018.*

Abstract: The scientific outcomes/novelties of the study should be highlighted. The presentation of the results could be more quantitative or precise.

*In a corrected version of the manuscript, we will improve the abstract as suggested by the reviewer to better highlight our scientific outcomes.*

Line 24: Add a comma after « namely ».
Line 29: Replace « A typical and prevalent approach" by "The reference method in soil physics".

Line 105: Please justify the choice of that case study with respect to the objectives of the study (and eventually the state of the art).

*At the start of our research, we conducted artificial infiltration experiments on an experimental platform, under controlled conditions: in sandy soils, with a known and fixed water table level, and a constant pressure head. Hence before applying the parameter estimation approach on the acquired data, we first wanted to perform a synthetic study under the same conditions to better understand the pertinence of the protocol in the context of parameter estimation.*

Line 106, "The infiltration is driven by a constant pressure head of 10 cm applied at the surface of the soil": Please justify the use of such boundary condition. Why not using a Neumann-type boundary condition (flux), as prevalent in environmental and agricultural applications? This would also be better suited to the use of GPR. It is worth noting that the antenna-medium coupling, which is permittivity dependent, will strongly affect the recorded radar waveforms for such experiment. This will inherently distort the estimation of the travel times if not modeled using a full solution of Maxwell's equations. This should be discussed in the presentation of the limitations of the present study.

*You're right that the Neumann-type boundary condition is often applied to model a natural flux entering a domain such as rainfall for example. However, to characterize the properties of porous media, different techniques can be investigated such as single or double ring infiltrometers. Our artificial infiltration experiment is inspired by such a field method in which a water head is applied to the domain surface. As it has been mentioned before, we performed several tests on an experimental platform. The main idea was to access more soil hydrodynamic*

*parameters (compared to conventional methods that mainly inform on $K_S$) using the GPR data monitored during the experiment.*

Figure 1: Which discretization was used to calculate the depth-dependent reflection coefficient? Please explain/justify.

*1D elements of 1 cm length were used, allowing a precise enough estimation of the wetting front position (maximum of the reflection coefficient) for the computation of the $TWT_f$ (travel time for the reflection on the wetting front) signal. Using finer elements would only increase the computation time, which we want as low as possible for the MCMC approach, without significantly modifying the computed $TWT_f$ signal.*

Line 124, "The initial condition is a hydrostatic pressure distribution corresponding to a water table at 100 cm depth": You can indicate that, in that case the soil moisture profile corresponds to the water retention curve.

*You are absolutely right that the hydrostatic condition means that the linear change in pressure between the surface and the water table will exactly match a part of the water retention curve. Notice that this is true for the initial condition and that the pressure and water content profiles will have a sharper shape as the infiltration front advances.*

Line 133: Add a comma before "respectively".

Line 138: Add a bracket before "Fig. 1".

Line 146: Use either "relative dielectric permittivity" or "dielectric constant", not "relative dielectric constant".

Line 157: Use italic for mathematical variable "N".

Equation (7): Please note that this assumes the medium electrical conductivity to be 0 (not true in practice but good approximation above about 300 MHz and below 1 GHz, as above 1 GHz dielectric losses do occur with water) and the relative magnetic permeability to be 1 (good approximation in most cases).

*You are right, it was indeed implicitly included in the reference to Annan (2003), but we will clearly state it in the corrected version of the manuscript. In fact, in our case we usually work with an 800 MHz antenna, low conductivity and not magnetic material.*

Line 169: "whatever the hydraulic parameters": Is it really true? The transition sharpness depends on the soil hydraulic properties and boundary conditions, as well as on soil type.

*Indeed, this statement is true in the test case considered, given the ranges of hydrodynamic parameter values that were investigated (Table 1), which will be clearly stated in a new version of the manuscript.*

Figure 2 caption: The value of the residual water content seems relatively large for a soil with such alpha and n values (sand-like). The choice of the soil hydraulic parameters should be justified in the text.

*We agree that this value is a little bit large but we can mention different studies (either dealing with parameter estimation based on experimental flow experiment or with numerical experiments) that have considered such values for the residual water content:*

*Haverkamp et al. (1977), Celia et al. (1991), Beydoun and Lehmann (2006), Younes et al. (2013). The main justification is that our experimental platform contains a sand media presenting this kind of values and even larger values. See for instance Dridi L (2006) p111.*

*Dridi, L. 2006. Transfert d'un mélange de solvants chlorés en aquifère poreux hétérogène: expérimentations sur site contrôlé et simulations numériques. Thèse de doctorat de l'Université Louis Pasteur de Strasbourg.*

*Also, according to Carsel and Parrish (1988), $\theta_r$ is $0.045 \pm 0.01$ in sands, so a value of 0.07 is a little bit large, but not so strange given the context of the investigated experimental platform.*

*Carsel, R. F. and R. S. Parrish (1988). "Developing joint probability distributions of soil water retention characteristics". In: Water Resources Research 24.5, pp. 755–769. DOI : 10.1029/WR024i005p00755.*

Equation (3): You may explain the choice of the exponent "1/2" in the hydraulic conductivity function. In principle, this exponent could take other values, depending on the soil type.

*Perfectly right; however, we adopt in this study the combination of Mualem and van Genuchten theories; hence the pore connectivity parameter is assumed to be 0.5.*

Line 205: Surround "respectively" with commas.

Table 1: Why choosing a relatively high value for the lower bound of n (1.5)? You may use 1.1 in order to include much more soils.

*We agree that n could decrease by a value slightly greater than 1 to include clay soils. The infiltration experiment on this type of soil would be physically longer and numerically more complicated. For convenience, we assume a lower bound set at 1.5; it would be possible to test 1.3 to include silty loam media.*

Line 265: Add a comma after "i.e.".

Line 289: "The parameter n has therefore a negligible effect on the TWT_f": Please explain the physical reason. Would the results be the same for different boundary conditions?

*The Mualem van Genuchten parameter n can have two counteractive effects. For the first effect, lower n values induce the expansion of the capillary transition zone (see Saintenoy and Hopmans, 2011), which increases the water content closer to the soil surface. This induces larger hydraulic conductivity values near the soil surface and hence faster wetting front propagation. On the other hand, lower n values also shrink the capillary fringe, which deepens the position where the wetting front collapses and reaches its steady state. Therefore, when n diminishes, the first effect increases the wetting front propagation speed, but the second effect lengthens the wetting front propagation path.*

*A "simple" Neumann boundary condition will probably not change the effect of n parameter on $TWT_f$. For real values of flux-type BC, the results might be different due to alternating inflows and outflows at the top surface, but this is not practical for the soil characterization.*

Line 316: Would that sensitivity be explained by the fact that at early times, the soil moisture profile does correspond to the soil water retention curve (hydrostatic equilibrium), which is significantly influenced by n? See also my previous comment.

*The initial hydrostatic profile has an effect on the propagation of the wetting front. We have tested scenarios by changing the depth of the water table which will lead to consider a larger range of the water retention curve. The simulation times are impacted but the conclusions remain unchanged.*

*Another possibility would have been to consider non-linear initial profiles with a drier zone on the first centimetres (situation that could be observed in the field). This belongs to further study, and is beyond the scope of this paper.*

Line 338: Please justify the use of a standard deviation of 0.5 ns.

*This corresponds to an uncertainty of 1 ns (2\*std), which is a realistic error given the data we work on and the GPR antenna of 800 MHz.*

The conclusions drawn in this paper are very specific to the theoretical case study that was analyzed. It would provide more value to the paper to include additional soil types, and/or, additional boundary conditions, and/or a real case study. At least discussions on the scope and limitations of the conclusions should be provided.

*We understand your comment in the case of a classical application of geophysical methods to determine hydrodynamic properties of the underground. We will better emphasize the idea that we improve a classical field characterization method (infiltrometry) by using GPR data. As suggested, we will improve the abstract and the introduction to complete the state-of-art and highlight our scope. As a result, the choice of the Dirichlet-type boundary condition is classical and relevant. We discussed the choice of the hydrostatic profile which does probably not change the results (we performed a few tests, see response to previous questions).*

*For the choice of the soil types, it is necessary to distinguish between the global sensitivity analysis and the parameter estimation parts of the manuscript. The Global sensitivity analysis (GSA) allows to consider a large panel of soil types (even if n is not so close to one, combination of parameters investigated by GSA is relatively exhaustive – see Table 1 for the intervals of the unsaturated soil parameters we cover). For the parameter estimation, we assumed Mualem van Genuchten parameters' values corresponding to the sand porous medium of our experimental platform.*

*We thank again the reviewer for his careful reading and its interesting comments that will help us to improve our manuscript.*

**Reviewer 2**

The manuscript titled "On the interest of ground penetrating radar data for the estimation of unsaturated soil parameters" written by Moua Rohianuu et al. presented the use of ground penetrating radar (GPR) time-lapse measurements for estimating hydrodynamic unsaturated soil parameters in synthetic infiltration experiments in which GPR travel time corresponds with different synthetic reflectors in the soil was used as observation measurements to estimate the soil physical parameters. Global sensitivity analysis was used to evaluate the sensitivity of soil model parameters and MCMC-based inversion method was used to estimate parameters and their associated confidence intervals. Below are my comments on this manuscript:

The manuscript was well written. However, the authors should clearly state the novelty in the study because using GPR travel time to estimate soil physical parameters are not new, especially with synthetic experiments.

*Thank you for your careful reading of our manuscript and your constructive comments. We agree to improve the abstract and introduction to better highlight the purpose of our study. Notice that the title will be modified to better focus on the subject, now:*
*"Coupled Hydrogeophysical inversion of an artificial infiltration experiment monitoring by ground penetrating radar: synthetic demonstration".*
*The state-of-art will also be enriched to underline our purpose which is to show the interest of GPR data and coupled hydrogeophysical model to better estimate the hydrodynamic Mualem van Genuchten parameters than classical field infiltrometry experiments.*

The synthetic experiments in this study were too ideal. We cannot find such cases in the reality. It's better if the synthetic experiments reflect the reality case. In this study, authors did not state the soil type used in their experiments. Because water dynamics in different soil types is different, I propose to perform the synthetic experiments with different soil types.
*The purpose is not to represent all real field conditions but rather to investigate efficiency of the GPR monitoring of an artificial infiltration, as a field characterization method.*
*Nonetheless, the Global Sensitivity Analysis we performed allows to investigate a large panel of porous media accordingly to the intervals defined in Table 1. We will better highlight this aspect in the new version of the manuscript.*
*Then the parameters estimation is demonstrated on a single synthetic case. Concerning the porous media used, we agree that the Mualem van Genuchten parameter estimation focuses on a sandy porous medium that is present in our experimental platform.*

Please briefly present the DREAM inversion algorithm in section 2. Why authors selected this algorithm for inversion?
*We will add details about DREAM at line 333. The full name of this algorithm is DiffeRential Evolution Adaptive Metropolis (DREAM); it samples the posterior probability density function (pdf) by running multiple Markov chains simultaneously for global exploration of the parameter space, and it automatically tunes the scale and orientation of the proposal distribution en route to the posterior target pdf. A MATLAB toolbox of the DREAM algorithm is available for Bayesian inference in fields ranging from physics, chemistry and engineering, to ecology, hydrology, and geophysics. See: Vrugt, J. A. 2016. Markov chain Monte Carlo simulation using the DREAM software package: Theory, concepts, and MATLAB implementation. Environmental Modelling & Software, 75, 273-316.*

At Line 145, please provide reference for permittivity of sand and the value of porosity. Porosity was fixed or considered to equal to saturated water content and change during inversion process?
*For the relative permittivity of the solid matrix, we used the value of 2.5 proposed by Léger et al. (2014). Yes, the porosity was set equal to the saturated water content parameter and thus also changed during the inversion process. This point will be indicated in the corrected version of the manuscript.*

In the global sensitivity analysis, did authors resize all parameter ranges to [0 1] before performing sensitivity analysis (because each parameter has its own unit and feasible range).
*Yes, we'll add this information in the new version of the manuscript.*

The manuscript should present the synthetic observation GPR travel time data for inversion in the 3 scenarios? How many GPR dataset were used for inversion? Did authors add noise to synthetic dataset? If not, noise should added to the synthetic data before performing inversion.

*As stated in line 337, a normally distributed noise with a standard deviation of 0.5 ns was added to the synthetic generated observations. The raw (no noise added) TWT synthetic signals are already represented in Fig. 2, as stated in line 339, and they are the same in every scenario investigated, but the added noise is always randomly generated and is therefore not always the same between the scenarios.*

Why the perfect solutions (inverted parameters equal to synthetic ones) did not obtained. In my opinion, if the perfect solutions were not obtained, the MCMC iteration did not reach the stable state. In addition, the manuscript should show the probability distribution of parameters for different scenarios, only present the confidence interval (CI) are not sufficient.

*The histograms represented are built with simulations where the chains are considered stable, i.e., where the Gelman Rubin criterion is verified (see Gelman and Rubin 1992) and the chains are not autocorrelated. We'll add this information in the new version of the manuscript.*

*In the scenarios where some estimated parameter values do not perfectly match the sought values, deviations do not come from an instability of the chains, but can rather be due to a low sensitivity of the concerned TWT data towards those parameters, in the sense that strong variations of the parameters will not affect the data significantly enough.*

*We'll also add the posterior distributions of three scenarios (1, 3 and 4) to further illustrate how combining the two types of TWT signal, instead of using them separately, can reduce the uncertainty.*

*Gelman, A.; Rubin, D.B. Inference from Iterative Simulation Using Multiple Sequences. Stat. Sci. 1992, 7, 457–472.*

Please explain the high correlation between Ks and $\theta_r$ and n and $\theta_r$. If this correlation influence the uncertainties of these parameters?

*As you mention (and it is also written in the manuscript), moderate correlations among Mualem van Genuchten hydraulic parameters appears in our results. It is not specific to our study and we can refer to Carsel and Parrish (1988) for instance. As you know, strong correlations between the optimized parameters indicate that the parameters cannot be well simultaneously estimated and that one of the parameters should be independently determined and fixed during the process. Slow convergence and non-uniqueness during the inversion process can be related to these correlations, which also increase parameter uncertainty. Our study shows that using different TWT signals is a good way to improve the quality of the parameter estimation.*

*Carsel, R. F., & Parrish, R. S. 1988. Developing joint probability distributions of soil water retention characteristics. Water resources research, 24(5), 755-769.*

In sensitivity analysis, parameter n is lowly sensitive with GPR travel time in the scenario 1 (Figure 4a) but well identified by MCMC inversion (Table 3). Please explain.

*We agree that this result is quite surprising; n is quite well identified since the sensitivity of n is negligible for scenario 1. The same observation also applies to the $\theta_S$ parameter.*

*Two elements to answer this interesting question. On the one hand, n values comprised between 1.5 and 3 could represent silty loam – sandy loam or sand. So, it's not obvious to consider this result as a good identification! We have to be careful. On the other hand, we have performed other estimations (not mentioned in the manuscript) with different target values of the Mualem van Genuchten parameters; a greater value of the "n" parameter (around 6 for instance) leads to a relatively good estimation (6.97) but the parameter is not well identified since its posterior interval remains large, though it is smaller than the prior parameter interval.*

*We will modify the manuscript to temper this result.*

*Finally, our conclusion remains relevant (for the different investigations we performed) namely that adding TWT signals (i.e., scenarios 4 and 5) helps further reducing the variation interval (see ratio in Table 3).*

---

## Author Response (AR1)

**Final Authors' Comments**

**Title:** On the interest of ground penetrating radar data for the estimation of unsaturated soil parameters
**Author(s):** Rohianuu Moua et al.
**MS No.:** egusphere-2022-936
**MS type:** Research article

Reviewers' comments are written in black.
*Authors' responses are written in italic blue.*
* * *
**Reviewer 1**

**General comments**

The authors analyze, through a numerical experiment, the retrieval of the unsaturated soil hydraulic properties from ground-penetrating radar (GPR) travel times corresponding to the wetting front as well as to fixed reflectors during an infiltration event. The hydrodynamic simulation involves a 1D solution of Richard's equation and the top and bottom boundary conditions are a 10 cm pressure head (Dirichlet) and a 1 m deep water table, respectively. Only one soil type, sand-like, is analyzed. The radar system reduces to a point at the soil surface, under water, and propagation times are calculated from propagation velocities derived from water content. The radar-antenna system and interactions with the medium are not accounted for. Sensitivity analyses as well as parameter estimation using the Markov Chain Monte Carlo Bayesian approach show that travel time information indeed provides valuable information to estimate the soil hydraulic properties. Different parameter sensitivities and corresponding uncertainties are observed and discussed.

The paper is well written and presented. It is technically sound and of interest to the scientific community. Nevertheless, compared to the state of the art, its novelty is quite limited. It is a case study and the interest mainly lies in the specific boundary conditions that are used for the hydrodynamic event as well as to the corresponding analyses. The fact that the radar and radar-medium interactions are not modeled limits the scope of the conclusions for real applications. The physical interpretation of the results could be deepened. The state of the art close to the topic of interest should be reviewed and links should be made with the observations of the authors.

*We thank the referee for his very constructive comments. In the revised version, we enlarged the state of the art, deepened the physical interpretation of the results and related them to the state of the art.*

**Specific comments**

Title and state of the art: The title should be more informative regarding the content of the paper. Indeed, the interest of ground penetrating radar to identify the soil hydraulic properties has already been demonstrated by many studies during these last two last decades. In that respect, a deeper literature review should be made to demonstrate the interest of the present study. Several studies relatively close to the topic of this paper are not referred to.

*We changed the title to:*

*"Coupled Hydrogeophysical inversion of an artificial infiltration experiment monitoring by ground penetrating radar: synthetic demonstration."*

*References that have been already cited in the old version, are now further discussed in the introduction section to emphasize the novelty of this research, such as: Saintenoy et al., 2008; Moysey, 2010; Scholer et al., 2011; Jadoon et al., 2012; Jaumann and Roth, 2018; Busch et al., 2013; Leger et al., 2014; Jonard et al., 2015; Leger et al., 2016; Leger et al., 2020*

Abstract: The scientific outcomes/novelties of the study should be highlighted. The presentation of the results could be more quantitative or precise.

*We improved the abstract as suggested by the reviewer to better highlight our scientific outcomes, and we also provided some quantitative results.*

Line 24: Add a comma after « namely ».
*Correction made.*

Line 29: Replace « A typical and prevalent approach" by "The reference method in soil physics".
*Correction made.*

Line 105: Please justify the choice of that case study with respect to the objectives of the study (and eventually the state of the art).

*At the start of our research, we conducted artificial infiltration experiments on an experimental platform, under controlled conditions: in sandy soils, with a known and fixed water table level, and a constant pressure head. Hence before applying the parameter estimation approach on the acquired data, we first wanted to perform a synthetic study under the same conditions to better understand the pertinence of the protocol in the context of parameter estimation.*

*These considerations are now mentioned (lines 163-165)*

Line 106, "The infiltration is driven by a constant pressure head of 10 cm applied at the surface of the soil": Please justify the use of such boundary condition. Why not using a Neumann-type boundary condition (flux), as prevalent in environmental and agricultural applications? This would also be better suited to the use of GPR. It is worth noting that the antenna-medium coupling, which is permittivity dependent, will strongly affect the recorded radar waveforms for such experiment. This will inherently distort the estimation of the travel times if not modeled using a full solution of Maxwell's equations. This should be discussed in the presentation of the limitations of the present study.

*You're right that the Neumann-type boundary condition is often applied to model a natural flux entering a domain such as rainfall for example. However, to characterize the properties of porous media, different techniques can be investigated such as single or double ring infiltrometers. Our artificial infiltration experiment is inspired by such a field method in which a water head is applied to the domain surface. As it has been mentioned before, we performed several tests on an experimental platform. The main idea was to access more soil hydrodynamic parameters (compared to conventional methods that mainly inform on $K_s$) using the GPR data*

*monitored during the experiment.*

*This is now mentioned in the manuscript (lines 167-170)*

Figure 1: Which discretization was used to calculate the depth-dependent reflection coefficient? Please explain/justify.

*1D elements of 1 cm length were used, allowing a precise enough estimation of the wetting front position (maximum of the reflection coefficient) for the computation of the $TWT_f$ (travel time for the reflection on the wetting front) signal. Using finer elements would only increase the computation time, which we want as low as possible for the MCMC approach, without significantly modifying the computed $TWT_f$ signal.*

*This is now mentioned in the manuscript (lines 245-246)*

Line 124, "The initial condition is a hydrostatic pressure distribution corresponding to a water table at 100 cm depth": You can indicate that, in that case the soil moisture profile corresponds to the water retention curve.

*It depends on what you call the water retention curve? Isn't it always the same as the soil moisture characteristic, at the hydrostatic equilibrium, after a precipitation event, or in dry conditions?*

Line 133: Add a comma before "respectively".*Correction made.*
Line 138: Add a bracket before "Fig. 1". *Correction made.*
Line 146: Use either "relative dielectric permittivity" or "dielectric constant", not "relative dielectric constant". *Correction made.*
Line 157: Use italic for mathematical variable "N". *Correction made.*

Equation (7): Please note that this assumes the medium electrical conductivity to be 0 (not true in practice but good approximation above about 300 MHz and below 1 GHz, as above 1 GHz dielectric losses do occur with water) and the relative magnetic permeability to be 1 (good approximation in most cases).

*You are right, it was indeed implicitly included in the reference to Annan (2003). It is now clearly stated in the manuscript (lines 215-216)*

*In fact, in our case we usually work with an 800 MHz antenna, low conductivity and not magnetic material.*

Line 169: "whatever the hydraulic parameters": Is it really true? The transition sharpness depends on the soil hydraulic properties and boundary conditions, as well as on soil type.

*Indeed, this statement is true in the test case considered, given the ranges of hydrodynamic parameter values that were investigated (Table 1).*

*It is now stated in the text as suggested here (lines 236-237)*

Figure 2 caption: The value of the residual water content seems relatively large for a soil with such alpha and n values (sand-like). The choice of the soil hydraulic parameters should be justified in the text.

*We agree that this value is a little bit large but we can mention different studies (either dealing with parameter estimation based on experimental flow experiment or with numerical experiments) that have considered such values for the residual water content:*

*Haverkamp et al. (1977), Celia et al. (1991), Beydoun and Lehmann (2006), Younes et al. (2013). The main justification is that our experimental platform contains a sand media presenting this kind of values and even larger values. See for instance Dridi L (2006) p111.*

*Dridi, L. 2006. Transfert d'un mélange de solvants chlorés en aquifère poreux hétérogène : expérimentations sur site contrôlé et simulations numériques. Thèse de doctorat de l'Université Louis Pasteur de Strasbourg.*

*Also, according to Carsel and Parrish (1988), $\theta_r$ is $0.045 \pm 0.01$ in sands, so a value of 0.07 is a little bit large, but not so strange given the context of the investigated experimental platform.*

*Carsel, R. F. and R. S. Parrish (1988). "Developing joint probability distributions of soil water retention characteristics". In: Water Resources Research 24.5, pp. 755–769. DOI : 10.1029/WR024i005p00755.*

*We cite these references only to try to answer the reviewers' remarks but, if necessary, they can be added to the manuscript's bibliography.*

Equation (3): You may explain the choice of the exponent "1/2" in the hydraulic conductivity function. In principle, this exponent could take other values, depending on the soil type.

*Perfectly right; however, we adopt in this study the combination of Mualem and van Genuchten theories; hence the pore connectivity parameter is assumed to be 0.5.*

*It is now mentioned in the text (line 199-200)*

Line 205: Surround "respectively" with commas. *Correction made.*

Table 1: Why choosing a relatively high value for the lower bound of n (1.5)? You may use 1.1 in order to include much more soils.

*We agree that n could decrease by a value slightly greater than 1 to include clay soils. The infiltration experiment on this type of soil would be physically longer and numerically more complicated.*

*We tested the GSA with a lower limit of 1.3 to include silty loam media. See results below:*

[Figure]

*The same observations are made, where values between 1.3 and 1.5 also have a strong contribution to the variance of the TWT signals.*

*For the parameter estimation, there is no point to increase the prior interval if the target value of n remains 2.68.*

*Hence for convenience, we can assume a lower bound set at 1.5.*

Line 265: Add a comma after "i.e.". *Correction made.*

Line 289: "The parameter n has therefore a negligible effect on the TWT_f": Please explain the physical reason. Would the results be the same for different boundary conditions?

*The Mualem van Genuchten parameter n can have two counteractive effects. For the first effect, lower n values induce the expansion of the capillary transition zone (see Saintenoy and Hopmans, 2011), which increases the water content closer to the soil surface. This induces larger hydraulic conductivity values near the soil surface and hence faster wetting front propagation. On the other hand, lower n values also shrink the capillary fringe, which deepens the position where the wetting front collapses and reaches its steady state. Therefore, when n diminishes, the first effect increases the wetting front propagation speed, but the second effect lengthens the wetting front propagation path.*

*A "simple" Neumann boundary condition will probably not change the effect of n parameter on $TWT_f$. For real values of flux-type BC, the results might be different due to alternating inflows and outflows at the top surface, but this is not practical for the soil characterization.*

Line 316: Would that sensitivity be explained by the fact that at early times, the soil moisture profile does correspond to the soil water retention curve (hydrostatic equilibrium), which is significantly influenced by n? See also my previous comment.

*The initial hydrostatic profile has an effect on the propagation of the wetting front. We have tested scenarios by changing the depth of the water table which will lead to consider a larger range of the water retention curve. The simulation times are impacted but the conclusions remain unchanged.*

*Another possibility would have been to consider non-linear initial profiles with a drier zone on the first centimeters (situation that could be observed in the field). This belongs to further study, and is beyond the scope of this paper.*

Line 338: Please justify the use of a standard deviation of 0.5 ns.

*This corresponds to an uncertainty of 1 ns (2\*std), which is a realistic error given the data we work on and the GPR antenna of 800 MHz.*
*This is now stated in the text (lines 420-421)*

The conclusions drawn in this paper are very specific to the theoretical case study that was analyzed. It would provide more value to the paper to include additional soil types, and/or, additional boundary conditions, and/or a real case study. At least discussions on the scope and limitations of the conclusions should be provided.

*"The conclusions drawn in this paper are very specific to the theoretical case study" but cover a very large range of soil types. The measuring protocol is specific because our goal is to improve a classical field characterization method (e.g., infiltrometry) by using GPR data. As suggested, we improved the abstract and the introduction to complete the state-of-art and highlight our scope. As a result of the measuring protocol, the choice of the Dirichlet-type boundary condition is classical and relevant. In the present response to the reviewer, we discussed the choice of the hydrostatic profile as initial condition.*

*For the choice of the soil types, it is necessary to distinguish between the global sensitivity analysis and the parameter estimation parts of the manuscript. The Global sensitivity analysis (GSA) allows to consider a large panel of soil types (even if n is not so close to one, combination of parameters investigated by GSA is relatively exhaustive – see Table 1 for the intervals of the unsaturated soil parameters we cover). For the parameter estimation, we assumed Mualem van Genuchten parameters' values corresponding to the sand porous medium of our experimental platform.*

*This is now mentioned in the manuscript (lines 289-291; lines 413-415)*

*We thank again the reviewer for his careful reading and its interesting comments that will help us to improve our manuscript.*

**Reviewer 2**

The manuscript titled "On the interest of ground penetrating radar data for the estimation of unsaturated soil parameters" written by Moua Rohianuu et al. presented the use of ground penetrating radar (GPR) time-lapse measurements for estimating hydrodynamic unsaturated soil parameters in synthetic infiltration experiments in which GPR travel time corresponds with different synthetic reflectors in the soil was used as observation measurements to estimate the soil physical parameters. Global sensitivity analysis was used to evaluate the sensitivity of soil model parameters and MCMC-based inversion method was used to estimate parameters and their associated confidence intervals. Below are my comments on this manuscript:

The manuscript was well written. However, the authors should clearly state the novelty in the study because using GPR travel time to estimate soil physical parameters are not new,especially with synthetic experiments.

*Thank you for your careful reading of our manuscript and your constructive comments. We now have improved the abstract and introduction to better highlight the purpose of our study. The title has been modified to better focus on the subject, now:*
*"Coupled Hydrogeophysical inversion of an artificial infiltration experiment monitoring by ground penetrating radar: synthetic demonstration".*
*The state-of-art has also been enriched to underline our purpose which is to show the interest of GPR data and coupled hydrogeophysical model to estimate, with a quick and non-destructive approach, the hydrodynamic Mualem van Genuchten parameters, better than with classical field infiltrometry experiments.*
*All these considerations are now mentioned in the text (abstract, introduction)*

The synthetic experiments in this study were too ideal. We cannot find such cases in the reality. It's better if the synthetic experiments reflect the reality case. In this study, authors did not state the soil type used in their experiments. Because water dynamics in different soil types is different, I propose to perform the synthetic experiments with different soil types.

*The purpose is not to represent all real field conditions but rather to investigate efficiency of a given protocol: the GPR monitoring of an artificial infiltration, as a field characterization method.*
*Nonetheless, the Global Sensitivity Analysis we performed allows to investigate a large panel of porous media accordingly to the intervals defined in Table 1. We better highlighted this aspect in the new version of the manuscript.*
*Then the parameters estimation performance is demonstrated on a single synthetic case. Concerning the porous media used for this single case, we agree that the Mualem van Genuchten parameter estimation focuses on a sandy porous medium that is favorable (both for GPR propagation and for the duration of the infiltration experiment), but is present in our local experimental platform.*
*These facts are mentioned in the improved manuscript (lines 289-291; lines 413-415)*

Please briefly present the DREAM inversion algorithm in section 2. Why authors selected this algorithm for inversion?
*We added details about DREAM at line 407.*

At Line 145, please provide reference for permittivity of sand and the value of porosity. Porosity was fixed or considered to equal to saturated water content and change during inversion process?

*For the relative permittivity of the solid matrix, we used the value of 2.5 proposed by Léger et al. (2014). Yes, the porosity was set equal to the saturated water content parameter and thus also changed during the inversion process. This point will be indicated in the corrected version of the manuscript.*
*These statements have been added in the manuscript (line 212)*

In the global sensitivity analysis, did authors resize all parameter ranges to [0 1] before performing sensitivity analysis (because each parameter has its own unit and feasible range). *yes, we normalized in the range [-1; 1]. We added this information at lines 297-298.*

The manuscript should present the synthetic observation GPR travel time data for inversion in the 3 scenarios? How many GPR dataset were used for inversion? Did authors add noise to synthetic dataset? If not, noise should added to the synthetic data before performing inversion.

*As stated in line 337, a normally distributed noise with a standard deviation of 0.5 ns was added to the synthetic generated observations. The raw (no noise added) TWT synthetic signals are already represented in Fig. 2, as stated in line 339, and they are the same in every scenario investigated, but the added noise is always randomly generated and is therefore not always the same between the scenarios.*

Why the perfect solutions (inverted parameters equal to synthetic ones) did not obtained. In my opinion, if the perfect solutions were not obtained, the MCMC iteration did not reach the stable state. In addition, the manuscript should show the probability distribution of parameters for different scenarios, only present the confidence interval (CI) are not sufficient.
*The histograms represented are built with simulations where the chains are considered stable, i.e., where the Gelman Rubin criterion is verified (see Gelman and Rubin 1992) and the chains are not autocorrelated.*
*This information is now added in the new version of the manuscript.*

*In the scenarios where some estimated parameters values do not perfectly match the sought values, deviations do not come from an instability of the chains, but can rather be due to a low sensitivity of the concerned TWT data to those parameters, in the sense that strong variations of the parameters will not affect the data significantly enough.*

*We also added the posterior distributions of three scenarios (1, 3 and 4) on Fig. 7 to further illustrate how combining the two types of TWT signal, instead of using them separately, can reduce the uncertainty.*

Please explain the high correlation between Ks and $\theta_r$ and n and $\theta_r$. If this correlation influence the uncertainties of these parameters?
*As you mention (and it is also written in the manuscript), moderate correlations among Mualem van Genuchten hydraulic parameters appears in our results. It is not specific to our study and we can refer to Carsel and Parrish (1988) for instance. Strong correlations between the optimized parameters indicate that the parameters cannot be well simultaneously estimated and that one of the parameters should be independently determined and fixed duringthe process. Slow convergence and non-uniqueness during the inversion process can be relatedto these correlations, which also increase parameter uncertainty. Our study shows that using different TWT signals is a good way to improve the quality of the parameter estimation.*
*This topic is not discussed in the manuscript, but we can mention it if necessary, and add the reference to Carsel and Parrish.*
*Carsel, R. F., & Parrish, R. S. 1988. Developing joint probability distributions of soil water retention characteristics. Water resources research, 24(5), 755-769.*

In sensitivity analysis, parameter n is lowly sensitive with GPR travel time in the scenario 1 (Figure 4a) but well identified by MCMC inversion (Table 3). Please explain.

*We agree that this result is quite surprising; n is quite well identified since the sensitivity of n is negligible for scenario 1. The same observation also applies to the $\theta_s$ parameter.*

*Two elements to answer this interesting question. On the one hand, n values comprised between 1.5 and 3 could represent silty loam – sandy loam or sand. So, it's not obvious to consider this result as a good identification! We have to be careful. On the other hand, we have performed other estimations (not mentioned in the manuscript) with different target values of the Mualem van Genuchten parameters; a greater value of the "n" parameter (around 6 for instance) leads to a relatively good estimation (6.97) but the parameter is not well identified since its posterior interval remains large, though it is smaller than the prior parameter interval.*

*This is now mentioned in the manuscript (lines 457-462)*

*Finally, our conclusion remains relevant (for the different investigations we performed) namely that adding TWT signals (i.e., scenarios 4 and 5) helps further reducing the variation interval (see ratio in Table 3).*

---

## Author Response (AR2)

First contact:
Rohianuu Moua
Corresponding author:
Rohianuu Moua
Handling editor:
Marnik Vanclooster

**Answers to Revision required 15 may 2023**

20 July 2023
Editor decision: Reconsider after major revisions (further review by editor and referees)
by Marnik Vanclooster

**Public justification (visible to the public if the article is accepted and published)**:
In their revised manuscript, the authors have made an effort to address the comments provided by the reviewers. While the revisions made are commendable, they primarily focus on minor improvements rather than addressing the major revisions suggested by the reviewers. The authors state in the abstract that the originality of their work lies in suggesting a statistical parameter estimation approach using Markov Chain Monte Carlo (MCMC) to obtain direct estimates of parameter uncertainties. However, it should be noted that parameter uncertainty estimation in inverse problems is a well-established theory and not fundamentally original. The true originality of this work lies primarily in the investigation of specific boundary conditions, as mentioned in the previous review. Unfortunately, the authors did not incorporate additional analyses with different soil types or other boundary conditions as suggested by both reviewers.

*Answer*

*In the revised version of the manuscript, we have now overcome this weakness of our study. We have applied the methodology to several types of sand, also considering different depths of the water table and various pressure heads at the top surface. These modifications are described from lines 501 to 521, also throughout table 4 and an additional figure (n°9). Our conclusions have been extended to add the main findings (L555 - 562)*

Despite these limitations, the manuscript has several strengths. The authors have effectively clarified and corrected various points, resulting in a well-written paper. The numerical experiment presented, which involves the retrieval of unsaturated soil hydraulic properties from ground-penetrating radar (GPR) travel times, is technically sound and of interest to the scientific community.
In the previous review, a summary of the manuscript, highlighting its main aspects was provided. The authors analyze the retrieval of unsaturated soil hydraulic properties using GPR travel times associated with the wetting front and fixed reflectors during an infiltration event. They employ a 1D solution of Richard's equation to simulate the hydrodynamics, with specific boundary conditions consisting of a 10 cm pressure head (Dirichlet) and a 1 m deep water table. The analysis focuses on a single soil type, resembling sand, and considers the radar system as a point at the soil surface, neglecting radar-antenna system interactions.

*Answer*

*Details have already been provided concerning the porous media considered and the boundary conditions (see L501-521 / 555-562). For radar-antenna system interaction, we have chosen to work on travel times to avoid such kind of problems (calibration of antenna amplitude, etc.).*

*The applicability and the portability of the procedure to other users is insured by the easiness of the protocol. And we remind that the specific geometry has been taken into account with the distance between TX and RX. Explanations have been included in the revised version on lines 229-239.*

Sensitivity analyses and parameter estimation using the MCMC Bayesian approach demonstrate the informative nature of travel time data for estimating soil hydraulic properties, revealing different parameter sensitivities and corresponding uncertainties. Although the manuscript is well written and presented, its novelty remains limited compared to the existing state of the art. It primarily serves as a case study with specific boundary conditions, making its interest predominantly centered around these conditions and corresponding analyses.

*Answer*

*Our approach consists in characterizing a soil using an infiltrometry test, which generally does not combine hydraulic and geophysical measurements. To reinforce the scope of the methodology, we have now considered several types of sand (whose van Genuchten hydrodynamic parameters correspond to those of an experimental site) and have also tested the impact of the the water table depth and the height of the ponding water applied.*

Furthermore, the lack of modeling for the radar system and its interactions with the medium restricts the practical applicability of the conclusions. To enhance the paper, the authors should delve deeper into the physical interpretation of the results and thoroughly review the literature related to their topic (e.g., not cited Jadoon et al., WRR, 2008, amongst several other key publications), drawing connections between their observations and the existing body of knowledge.

*Answer*

*Additional simulations have been performed to better support the conclusions reached and show that the methodology is effective. References have been added to improve the state of the art (Jadoon et al. 2008, Tran et al. 2014) and the interpretation of results has also been refined (L363-364 / 392-397).*

Report #1

In their revised manuscript, the authors have made an effort to address the comments provided by both myself and another reviewer. While the revisions made are commendable, they primarily focus on minor improvements rather than addressing the major revisions suggested by the reviewers. The authors state in the abstract that the originality of their work lies in suggesting a statistical parameter estimation approach using Markov Chain Monte Carlo (MCMC) to obtain direct estimates of parameter uncertainties. However, it should be noted that parameter uncertainty estimation in inverse problems is a well-established theory and not fundamentally original.

The true originality of this work lies primarily in the investigation of specific boundary conditions, as mentioned in my previous review. Unfortunately, the authors did not incorporate additional analyses with different soil types or other boundary conditions as suggested by both reviewers. As a result, the outcomes of their analyses are of limited general interest.

Despite these limitations, the manuscript has several strengths. The authors have effectively clarified and corrected various points, resulting in a well-written paper. The numerical experiment presented, which involves the retrieval of unsaturated soil hydraulic properties from ground-penetrating radar (GPR) travel times, is technically sound and of interest to the scientific community.

In my previous review, I provided a summary of the manuscript, highlighting its main aspects. The authors analyze the retrieval of unsaturated soil hydraulic properties using GPR travel times associated with the wetting front and fixed reflectors during an infiltration event. They employ a 1D solution of Richard's equation to simulate the hydrodynamics, with specific boundary conditions consisting of a 10 cm pressure head (Dirichlet) and a 1 m deep water table. The analysis focuses on a single soil type, resembling sand, and considers the radar system as a point at the soil surface, neglecting radar-antenna system interactions. Sensitivity analyses and parameter estimation using the MCMC Bayesian approach demonstrate the informative nature of travel time data for estimating soil hydraulic properties, revealing different parameter sensitivities and corresponding uncertainties.

Although the manuscript is well written and presented, its novelty remains limited compared to the existing state of the art. It primarily serves as a case study with specific boundary conditions, making its interest predominantly centered around these conditions and corresponding analyses. Furthermore, the lack of modeling for the radar system and its interactions with the medium restricts the practical applicability of the conclusions. To enhance the paper, the authors should delve deeper into the physical interpretation of the results and thoroughly review the literature related to their topic (e.g., not cited Jadoon et al., WRR, 2008, amongst several other key publications), drawing connections between their observations and the existing body of knowledge.

*Answer*

*We thank the referee for his very constructive comments. We are well aware that certain points are prohibitive for publication in the egusphere journal, so we have addressed all your comments and improved the quality of our study.*

*So, following the initial improvements made, we have addressed the various shortcomings as follows. New simulations have now been carried out, considering several types of soil, studying the impact of the depth of the water table on parameter estimation, and also applying different water heights of surface water to be infiltrated from the infiltrometer device. Consequently, interpretations of our results have been enhanced to go deeper in the physics (L364-365 / 393-398). The reviewer can check the modifications in the revised manuscript from lines 501 to 521, also throughout table 4 and an additional figure (n°9). Our conclusions have been extended to add the main findings (L555 - 562). Finally, we also added the mentioned references to better draw the connection with existing literature on the subject and to reinforce the state of the art which had already been improved in the previous feedback.*